# Transforming Weather Data from Pixel to Latent Space

**Sijie Zhao** [1 2]  **Feng Liu** [2 3]  **Xueliang Zhang** [1]  **Hao Chen** [2]  **Tao Han** [2]  **Junchao Gong** [2 3]  **Ran Tao** [2]
**Pengfeng Xiao** [1]  **Xinyu Gu** [2]  **Lei Bai** [2]

## Abstract

The increasing impact of climate change and extreme weather events has spurred growing interest in deep learning for weather research. However, existing studies often rely on weather data in pixel space, which presents several challenges such as smooth outputs in model outputs, limited applicability to a single pressure-variable subset (PVS), and high data storage and computational costs. To address these challenges, we propose a novel Weather Latent Autoencoder (WLA) that transforms weather data from pixel space to latent space, enabling efficient data representation. By decoupling weather reconstruction from downstream tasks, WLA improves the accuracy and sharpness of weather task model results. The incorporated Pressure-Variable Unified Module transforms multiple PVS into a unified representation, enhancing the adaptability of the model in multiple weather scenarios. Furthermore, weather tasks can be performed in a low-storage latent space of WLA rather than a high-storage pixel space, thus significantly reducing data storage and computational costs. Through extensive experimentation, we demonstrate its superior compression and reconstruction performance, enabling the creation of the ERA5-Latent dataset with unified representations of multiple PVS from ERA5 data. The compressed full PVS in the ERA5-Latent dataset reduces the original 244.34 TB of data to 0.43 TB. The downstream task further demonstrates that task models can be applied to multiple PVS with low data costs in the latent space, achieving superior performance compared to their pixel-space counterparts.

[1]Nanjing University [2]Shanghai Artificial Intelligence Laboratory [3]Shanghai Jiao Tong University. Correspondence to: Xueliang Zhang <zxl@nju.edu.cn>, Hao Chen <chenhao1@pjlab.org.cn>.

*Proceedings of the 43rd International Conference on Machine Learning*, Seoul, South Korea. PMLR 306, 2026. Copyright 2026 by the author(s).

## 1. Introduction

The profound impact of climate change and extreme weather events on the Earth has attracted widespread attention (Patz et al., 2005; Wild et al., 2025; Chen et al., 2025). Recently, deep learning methods have made groundbreaking advancements in meteorology, leading to increasing interest in their application to weather research (Ravuri et al., 2021; LIU et al., 2022; Yang et al., 2023; Zhang et al., 2023b; Gong et al., 2024a;b). However, most existing studies focus primarily on weather-related tasks in the pixel space of weather data (Bi et al., 2023; Chen et al., 2023b;a). The efficiency of weather models in prior studies is often hindered by the inherent uncertainty of tasks and the diversity of data, whereas data costs are inflated by expensive storage and processing requirements.

Specifically, performing weather-related tasks in the pixel space presents three main limitations (as shown in Fig.1): 1) **Smooth Model Results**. Weather data contain rich small-scale extreme values. When performing tasks such as weather forecasting and downscaling in the pixel space, the model also needs to perform weather reconstruction, requiring a fine reconstruction of small-scale extreme values. However, the inherent uncertainty in weather-related tasks degrades the performance of small-scale extreme values reconstruction and extreme events prediction, leading to smooth results (Ravuri et al., 2021). 2) **Limited Model Applicability to a Single Pressure-Variable Subset (PVS)**. Weather data typically record various weather variables across multiple pressure levels, leading to significant data diversity in the pixel space (Astruc et al., 2024; Xiong et al., 2024). Different weather-related tasks and applications often require distinct PVS selections. For instance, the 500 hPa geopotential height and the 850 hPa wind fields are key to typhoon path prediction (Hua & Chong-Yin, 2010; Moore & Dixon, 2015). Conversely, the 500 hPa geopotential height, 700 hPa vertical velocity, and 925 hPa specific humidity serve as essential parameters for short-term rainfall forecasting (Kuligowski & Barros, 1998; Tian et al., 2015). However, models trained in pixel space are typically restricted to a single PVS, limiting their adaptability across multiple weather scenarios requiring different PVS compositions. 3) **High Data Storage and Computational Costs**. Pixel-based weather datasets can reach hundreds of terabytes (TB) or even petabytes (PB), leading to significant storage and computational costs (Hersbach et al.,

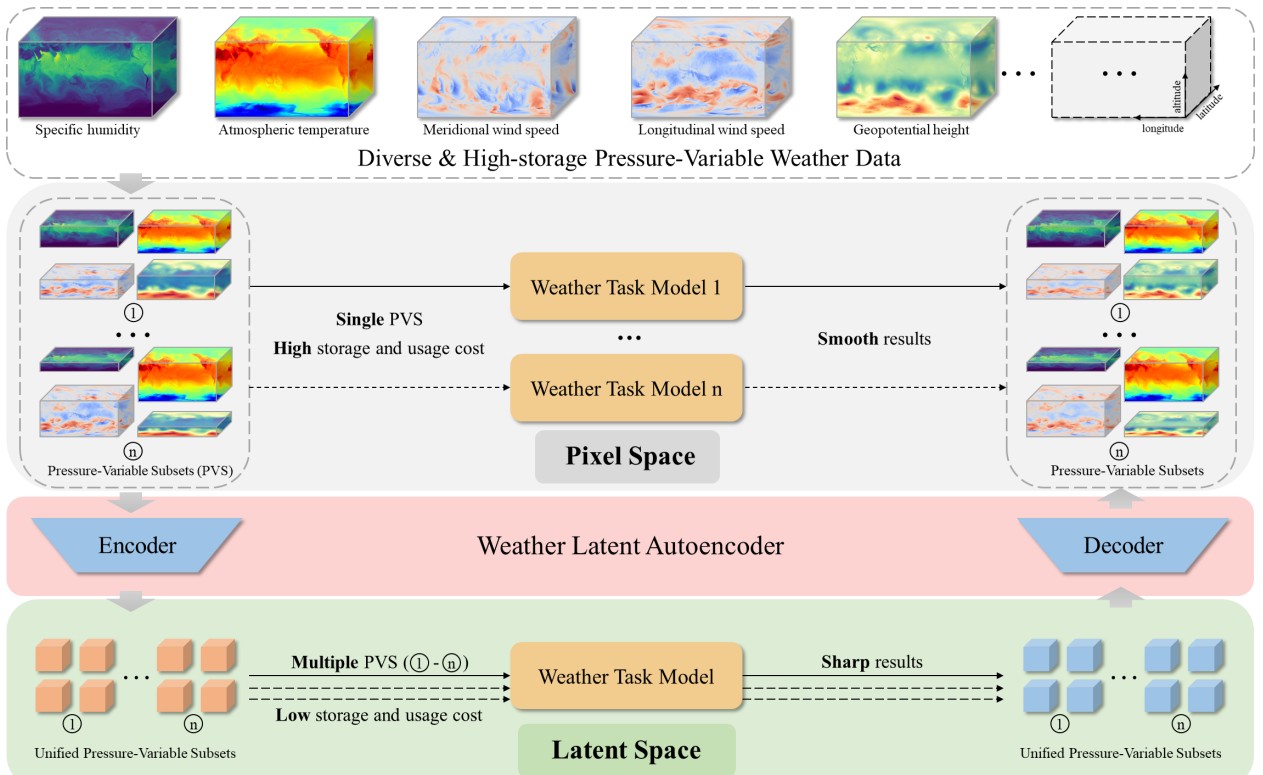

*Figure 1.* Transforming weather data from diverse and high-storage pixel space to unified and low-storage latent space for weather tasks using weather latent autoencoder. The weather task model in pixel space suffers from high data storage and computational costs and limited applicability to single pressure-variable subset, often yielding ambiguous results. In contrast, the model in latent space benefits from reduced data storage and computational costs, enabling the use of multiple pressure-variable subsets and producing sharper results.

2020). This poses a substantial challenge for the large-scale application of deep learning in meteorology (Klöwer et al., 2021; Han et al., 2024a).

To address the above limitations, we propose a novel approach that transforms weather data from pixel space to latent space for weather-related tasks. Specifically, we introduce the Weather Latent Autoencoder (WLA), as illustrated in Fig.1. WLA effectively encodes diverse and high-storage weather data from the pixel space to a unified and lower-storage latent space, facilitating its application to multiple PVSs. This transformation allows weather-task models to operate directly in the latent space, eliminating the need for pixel-space data, thereby enhancing their adaptability to different PVS compositions while significantly reducing data storage and computational costs.

Specifically, WLA addresses the aforementioned issues in three ways: 1) **Decoupling Weather Reconstruction from Weather Tasks**. In this approach, weather tasks are performed in the latent space, while weather reconstruction occurs within the pretrained Weather Latent Autoencoder. The pretrained WLA ensures that latent features effectively pre-

serve small-scale extreme values, allowing for high-quality reconstruction from these features. During weather tasks in the latent space, the uncertainty inherent in these tasks has minimal impact on the small-scale extreme values reconstruction of WLA, resulting in sharp and accurate outcomes for the weather task model. 2) **Unified Pressure-Variable Representation**. We introduce a Pressure-Variable Unified Module (PVUM) designed to transform any pressure-variable subset to a unified space. PVUM leverages pressure-variable metadata in weather data to generate adaptive mapping network weights through a hypernetwork, enabling the conversion of weather data from pixel space into a unified latent space. This framework allows the weather task model to seamlessly accommodate various types of weather data inputs in the latent space, enhancing its applicability across diverse weather scenarios. 3) **Latent Space Framework**. We propose the Latent Space Framework, which transitions weather task models from pixel space to latent space, significantly reducing data storage and computational costs. Thanks to WLA's superior compression and reconstruction capabilities, the latent data retain most of the information from the original pixel data, but with a much smaller storage

footprint. This results in a substantial reduction in storage costs. Furthermore, tasks such as model training, validation, and testing, which typically require large amounts of data, can be carried out using low-storage latent data, yielding significant savings in data computational costs.

To facilitate research on weather tasks performed directly in latent space, we introduce **ERA5-Latent**, a novel dataset derived from ERA5 (Hersbach et al., 2020). While raw ERA5 data offers high fidelity, its sheer size (hundreds of TB) is prohibitive. Addressing these challenges, we utilize our proposed WLA to transform high-resolution ERA5 data (721×1440 size) from pixel space into a compact latent representation. This ERA5-Latent dataset substantially reduces data costs and enables research using the full scope of ERA5 maps and diverse variable sets within the latent domain.

The original ERA5 data includes 164 variables and totals 244.34 TB. Our WLA transformation compresses this into a latent representation requiring only 0.43 TB, achieving a 566.3× compression ratio and significantly lowering storage costs. To support diverse modeling needs, ERA5-Latent offers unified latent representations for commonly used configurations: 6 upper-air variables across 6, 13, and 25 pressure levels; surface variables in sets of 4 and 8; and precipitation variables in sets of 1 and 6. Models can leverage this low-storage latent data for training, validation, and testing across various scenarios in latent space, minimizing data and computational expenses.

In summary, our main contributions are as follows:

1. We propose a novel framework that transforms weather data from pixel space to latent space for weather tasks, which enables sharper model results and unified weather representation, with significant reduction of data storage and computational costs.

2. We design the Weather Latent Autoencoder for the pixel-to-latent transformation of weather data. WLA can effectively transform any pressure-variable subset from pixel space to a unified latent space, providing excellent compression and reconstruction performance.

3. We have constructed the ERA5-Latent dataset, which provides large-scale ERA5 weather data with multiple pressure-variable subsets in a smaller data storage footprint and unified latent space.

## 2. Related Work

**Weather Data Compression**

Weather data compression has advanced from traditional linear quantization (GRIB2-based 17× compression of CAM (Inness et al., 2019; Klöwer et al., 2021)) to neural representation learning. Autoencoder-based models (Liang et al., 2023) and coordinate-aware networks (Huang & Hoefler, 2023) achieve high compression ratios through instance-specific overfitting, though often at the cost of generalization. Meta-learning methods like COIN++ (Dupont et al., 2022) address this by leveraging shared priors for modality-agnostic compression. More recent advances combine probabilistic modeling with entropy coding. For example, Mirowski et al. (Mirowski et al., 2024) achieve 1000× compression using hyperpriors and vector quantization, while CRA5 (Han et al., 2024a) employs a dual-variational transformer to optimize rate-distortion via hierarchical latent space modeling. These compression methods have limitations with the compressed data: either online decompression leads to computational resource consumption, or offline decompression leads to storage resource consumption. Our framework avoids this by directly using latent space data, reducing data calculation costs.

Furthermore, recent weather and Earth system foundation models leverage large-scale pre-training to map meteorological variables into compact, informative feature representations, which implicitly facilitates data reduction (Schmude et al., 2024; Bodnar et al., 2025; Price et al., 2025). Notably, GenCast (Price et al., 2025) mitigates the issue of over-smoothed outputs by employing probabilistic modeling, thereby capturing sharper and more realistic meteorological details. However, a fundamental distinction exists between these task-oriented representation models and dedicated data compression frameworks. Foundation models prioritize the embedding quality for downstream tasks, whereas weather compression models focus on unified representation and high compression ratios. Consequently, while foundation models excel in downstream task performance, WLA leads in data compression efficiency. Crucially, these two paradigms are complementary: weather foundation models can be constructed directly within WLA's latent space, enabling them to adapt seamlessly to diverse variable subsets without relying on independent linear layers for each variable as in Aurora (Bodnar et al., 2025).

**Managing the Diversity of Weather Data**

Earth science modeling is challenged by heterogeneous observational data. Current methods either rely on specialized architectures such as Omnisat's modality-specific encoders (Astruc et al., 2024) or on metadata-driven adaptation, as seen in DOFA's spectral self-supervision (Xiong et al., 2024). In weather forecasting, the combinatorial complexity of atmospheric variables and pressure levels often results in brittle models. For instance, FengWu (Chen et al., 2023a) employs 5 upper-air variables at 37 pressure levels with 4 surface variables, while Pangu (Bi et al., 2023) and FengWu-GHR (Han et al., 2024b) use 13 pressure levels for similar variables.

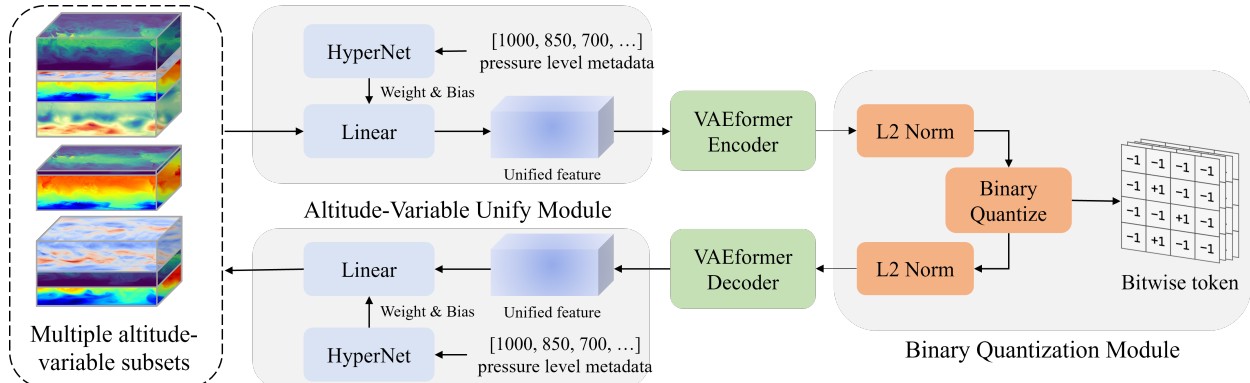

*Figure 2.* Architecture of the Weather Latent Autoencoder, which compresses weather data from a diverse, high-storage pixel space into a unified, low-storage latent space, and reconstructs it back into the pixel space.

## Low-Cost Weather Datasets

The exponential growth of weather data poses significant challenges in storage, computation, and accessibility. Curated low-cost datasets such as Weatherbench (Rasp et al., 2020) mitigate these issues by downsampling data and selecting pressure level subsets. In contrast, CRA5 retains ERA5's full 0.25° resolution across 159 fields, achieving high storage efficiency through neural compression at the expense of requiring decoder reconstruction. Collectively, these studies highlight the importance of efficient data representation.

## 3. Method

### 3.1. Overview of Weather Latent Autoencoder

The Weather Latent Autoencoder transforms weather data from diverse and high-storage pixel space into unified and low-storage latent space. As illustrated in Figure 2, our framework integrates three core components: (1) a Pressure-Variable Unified Module that leverages metadata information to align heterogeneous PVS features, (2) a VAEformer Encoder-Decoder pair adopting the transformer architecture from CRA5's pretraining stage (Han et al., 2024a) for latent feature compression/reconstruction, and (3) a Binary Quantization Module (BQM) that generates compact bitwise tokens through spherical normalization and binary quantization.

During the compression phase, the model starts with selecting multiple PVSs from a multiple pressure-variables weather dataset. The PVUM first converts pressure-variable metadata into adaptive parameters through a hypernetwork, enabling cross-scale feature alignment across disparate PVS. These unified features are subsequently encoded by the VAE-former encoder into low-dimensional latent representations, preserving essential weather patterns while discarding pixel-space redundancies. The BQM then projects the latent fea-

tures onto a unit spherical space through L2-normalization and applies binary quantization to produce storage-efficient bitwise tokens. This compression effectively reduces data storage compared to original pixel-space PVS representations.

The reconstruction phase executes an inverse transformation through three cascaded operations. Initially, bitwise tokens are mapped back to spherical space via L2-normalization. The VAEformer Decoder subsequently reconstructs unified features through upsampling operators, ensuring high-quality weather data reconstruction. Finally, the PVUM regenerates the original PVS by applying metadata-guided inverse transformations, thereby completing the latent-to-pixel space transformation cycle.

The latent space framework of WLA offers three fundamental benefits. First, the unified encoding enables weather-task models to directly operate on a unified latent space, eliminating structural modifications for cross-PVS generalization. Second, the WLA decouples weather reconstruction from task modeling. The uncertainty present in the weather tasks does not affect the weather reconstruction, ensuring that the task model can output sharp and accurate results. Third, data storage and computational costs are significantly reduced as model training, validation, and inference primarily utilize low-storage latent features, restricting pixel-space operations to final metric evaluation phases.

### 3.2. Pressure-Variable Unified Module

To map any pressure-variable subset from pixel space to a unified feature, we designed the Pressure-variable Unified Module, which utilizes the metadata of the pressure-variable subset to generate adaptive weights and biases for a linear layer, thereby enabling adaptive feature mapping.

As shown in Figure 3, given an input PVS tensor $X \in \mathbb{R}^{C_1 \times H \times W}$ with its pressure-variable metadata $M \in \mathbb{R}^{C_1}$ (where $C_1$ varies across tasks and scenarios), PVUM gener-

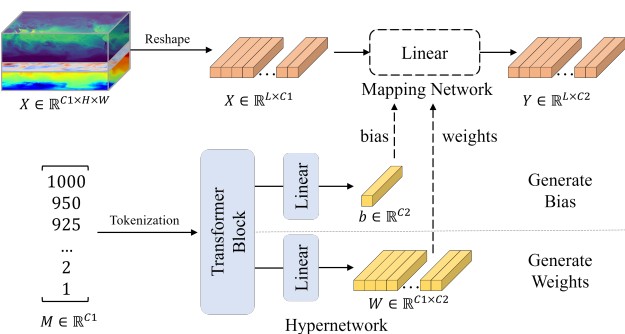

*Figure 3.* Workflow of Pressure-Variable Unified Module, which transforms diverse weather data into unified representation.

ates a unified feature $Y$ with fixed dimensionality through hypernetwork-based parameter generation. This process consists of three core operations: **Metadata Embedding:** The variable metadata $M$ containing physical attributes (pressure levels and variables) undergoes positional encoding followed by tokenization. A learnable class token [CLS] is prepended to the token sequence $T \in \mathbb{R}^{(C_1+1) \times d}$, where $d$ is the embedding dimension. **Cross-Variable Relation Modeling:** The token sequence passes through several transformer blocks for learning the relationships between the metadata. **Adaptive Parameter Generation:** The [CLS] token produces bias parameters $b \in \mathbb{R}^{C_2}$ via a linear projection, while the remaining tokens generate a weight matrix $W \in \mathbb{R}^{C_1 \times C_2}$ through another linear layer. The resulting weights $W$ and bias $b$ form a linear layer that maps the features with $C_1$ channels to features with $C_2$ channels. Therefore, the input $X$ is reshaped from $(C_1, H, W)$ to $(L, C_1)$, where $L = H \times W$, and then mapped to the target feature $Y$ with shape $(L, C_2)$ using the generated linear layer.

Due to the continuity, smoothness, and vertical mixing of the atmosphere, there is inherent similarity between different weather pressure levels and variable data, especially between adjacent pressure levels for the same variable (Zhang et al., 2023a). The hypernetwork of PVUM learns this relationship when modeling the metadata, allowing it to map similar weather variables and adjacent pressure levels to similar unified features.

### 3.3. Binary Quantization Module

To effectively compress weather features while preserving critical information, we propose the Binary Quantization Module that establishes a bi-directional mapping between continuous features and discrete binary tokens. As shown in Figure 2 (right), the module inherits the vector quantization framework from BSQ (Zhao et al., 2024) which has two key components: (1) spherical space projection for stable entropy loss estimation, and (2) deterministic binary quantization for hardware-friendly storage. The quantization pro-

cess consists of three stages: First, input features undergo L2 normalization to project them onto a spherical space, which not only stabilizes the subsequent quantization but also enables computation of the entropy loss with acceptable memory/space cost (Han et al., 2025). Second, we apply sign-based binary quantization where positive values are mapped to 1 and negative values to -1, generating compact bitwise tokens. During reconstruction, the bitwise tokens are inversely projected to the spherical space through L2 normalization before being fed to the VAEformer decoder for upsampling.

The compression ratio of our weather latent autoencoder can be formally analyzed through the data storage. Let the input feature tensor $F \in \mathbb{R}^{C \times H \times W}$ with float32 representation be compressed into binary tokens $B \in \{-1, 1\}^{C' \times H' \times W'}$. The spatial downsampling factors $(P_h, P_w) = (H/H', W/W')$ combined with channel dimension adjustment yield a compression ratio

$$R = \frac{C \cdot H \cdot W \cdot 32}{C' \cdot H' \cdot W'} = \frac{C}{C'} \cdot 32 \cdot P_h \cdot P_w. \quad (1)$$

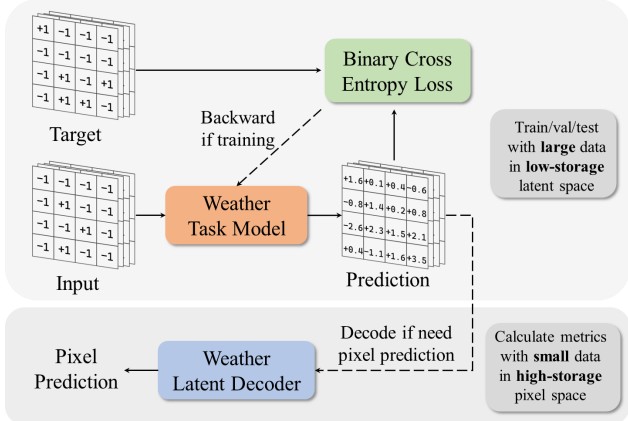

*Figure 4.* Overview of the Latent Space Framework.

### 3.4. Latent Space Framework

To minimize storage and computational overhead in deep learning-based weather modeling, we propose the Latent Space Framework (LSF), as illustrated in Figure 4. Leveraging a Weather Latent Autoencoder, LSF is grounded in two insights: latent representations preserve essential features, and data similarity remains consistent between pixel and latent spaces. Consequently, data-intensive phases are conducted entirely within the low-storage latent space. Downstream task models process latent tokens directly, with model training and performance evaluation using latent-space loss. High-resolution pixel space is reserved only for low-data tasks, such as metric calculations and result visualizations. By partitioning workflows based on data volume, LSF significantly reduces the data and computational costs.

*Table 1.* Compression result of WLA and several state-of-the-art compression methods. Weighted RMSE is utilized to quantify the reconstruction error of the compression model.

| METHOD | WEIGHTED RMSE ↓ | | | | | | | COMP. RATIO ↑ | BPSP ↓ |
|---|---|---|---|---|---|---|---|---|---|
| | UPPER-AIR VARIABLES | | | | SURFACE VARIABLES | | PRECIPITATION | | |
| | w500 | w700 | Q700 | Q1000 | TCC | SP | TP6H | | |
| VAR. STD (REF.) | 0.218 | 0.240 | 0.0025 | 0.0059 | 0.36 | 9584.49 | 1.57 | – | – |
| ELIC (HE ET AL., 2022) | 0.197 | 0.233 | 0.00076 | 0.00087 | 0.18 | 537.82 | 1.19 | 648.3 | 0.112 |
| IEN (XIE ET AL., 2021) | 0.213 | 0.247 | 0.00084 | 0.00092 | 0.23 | 688.27 | 1.03 | 202.5 | 0.158 |
| VQVAE (MIROWSKI ET AL., 2024) | 0.382 | 0.401 | 0.00108 | 0.00113 | 0.19 | 673.32 | 1.29 | **1100.0** | **0.029** |
| VQGAN (MIROWSKI ET AL., 2024) | 0.367 | 0.371 | 0.00101 | 0.00107 | 0.18 | 652.38 | 1.20 | **1100.0** | **0.029** |
| VAEFORMER (HAN ET AL., 2024A) | 0.117 | 0.134 | 0.00031 | 0.00035 | 0.12 | 376.90 | 0.80 | 323.1 | 0.099 |
| $WLA_{small}$ | **0.126** | **0.168** | **0.00038** | **0.00041** | **0.087** | **328.4** | **0.64** | 1251.7 | 0.025 |
| **WLA** | **0.076** | **0.083** | **0.00027** | **0.00028** | **0.055** | **257.88** | **0.47** | 625.9 | 0.051 |

## 4. Experimental Results

### 4.1. Original Dataset

The ERA5 dataset (Hersbach et al., 2020) is served as a standard for evaluating the Weather Latent Autoencoder in comparison to other models. To meet various weather-related needs, we organized three categories of variables: upper-air, surface, and precipitation variables. Considering the physical characteristics of different weather variables, we treat them as different modalities and designed distinct WLA architectures (Chen et al., 2023a). Meanwhile, a $WLA_{small}$ model with a smaller codebook size of $2^{64}$ was introduced to evaluate the performance of WLA against other methods under higher compression ratios. Experimental details can be found in Section A.1.

### 4.2. Overall Results

To demonstrate the effectiveness of the WLA in data compression, we compared it with several state-of-the-art compression methods (Elic (He et al., 2022), IEN (Xie et al., 2021), VQVAE (van den Oord et al., 2017), VAGAN (Esser et al., 2021), VAEformer (Han et al., 2024a)) across three metrics: compression ratio, bits per sub-pixel (bpsp) (Mentzer et al., 2019), and weighted RMSE (Han et al., 2024b) on representative upper-air, surface, and precipitation variables. Setting details can be found in Section A.2.

As shown in Table 1, WLA achieves superior overall compression performance across upper-air, surface, and precipitation variables compared to existing methods, characterized by higher compression ratios, lower bpsp values, and competitive weighted RMSE scores. This demonstrates that WLA effectively balances weather data compression with reconstruction. Notably, WLA demonstrates remarkable flexibility and versatility, enabling seamless adaptation to diverse variable combinations and complex application scenarios. The visualization results and information loss analysis of WLA can be found in Section A.7. The results show that WLA are highly effective at preserving the fine-grained, small-scale extreme values that are vital for meteorological applications.

### 4.3. Out-of-domain Generalization

To validate the generalization capability of WLA on unseen pressure levels and out-of-domain data, we conducted experiments using two distinct datasets. The first is the ERA5 dataset, which includes 37 pressure levels, of which only 25 were used during the training phase. The second is the HRES dataset, which was downsampled to match the spatial resolution of the ERA5 data.

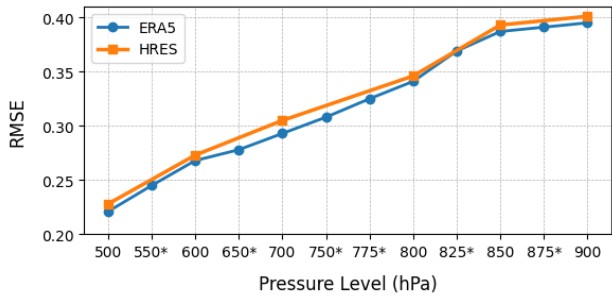

*Figure 5.* Generalization performance on unseen pressure level and out-of-domain data.

The experimental results in Fig.5 show the performance of WLA on the temperature variable across 12 pressure levels. These results indicate that WLA achieves robust reconstruction performance on pressure levels for which it was not trained (denoted by *). Furthermore, the model shows a reconstruction performance on the HRES data that is comparable to its performance on the ERA5 data. This demonstrates that WLA possesses strong generalization capabilities, both for unseen pressure levels and for out-of-domain data.

### 4.4. Derived ERA5-Latent Dataset

Leveraging the excellent compression and reconstruction performance of the Weather Latent Autoencoder, we transformed the multiple PVSs of ERA5 data into unified latent space, yielding the ERA5-Latent dataset. By utilizing the high compression rate of WLA, the ERA5-Latent dataset reduces the original 244.34 TB of data down to 0.43 TB, while providing a unified representation for multiple PVS. Dataset details can be found in subsection A.3.1.

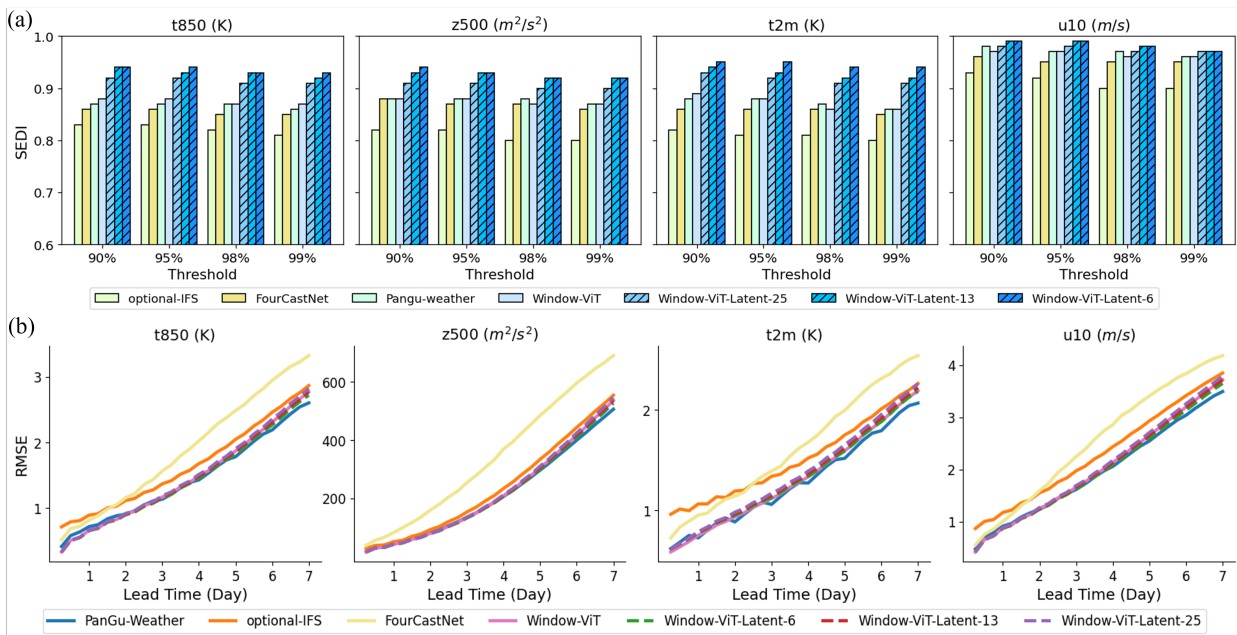

*Figure 6.* Comparison on the weather forecasting task. (a) Extreme weather forecasting performance on the SEDI metric. (b) RMSE for weather forecasting over a 7-day period. The number following "Window-ViT-latent" indicates the number of input pressure levels.

## 4.5. Downstream Task Evaluation

### 4.5.1. EVALUATION ON WEATHER FORECASTING

To demonstrate that models operating in a unified latent space can adapt to multiple PVS and generate sharper results compared to pixel-space models, we conducted experiments on the weather forecasting task. We used two models to conduct weather forecasting: the Window-ViT (WT) in the pixel space, and the Window-ViT-Latent (WTL) in the latent space. The experiments were performed on the ERA5 dataset and the ERA5-Latent dataset. Specifically, following the settings in (Bi et al., 2023), we conduct 7-day weather forecasting at 6-hour intervals. For the WT, we use 13 pressure levels for five upper-air variables along with four surface variables. In contrast, to evaluate the WTL's adaptability to multiple PVS, we experiment with multiple pressure levels (25, 13, 6) for the upper-air variables for the latent model. This experiment takes the same configuration as the state-of-the-art model (Chen et al., 2023a).

To comprehensively evaluate performance on extreme events, we employed two distinct metrics: the Symmetric Extremal Dependency Index (SEDI) (Kurth et al., 2023; Xu et al., 2024) to assess event detection capability, and the Relative Quantile Error (RQE) (Kurth et al., 2023) to evaluate the preservation of extreme event magnitudes. We benchmarked our models against the physics-based Operational Integrated Forecasting System (IFS) (Bougeault et al., 2010) and the AI-based Pangu-Weather (Bi et al., 2023). Experimental details are provided in Section A.3.

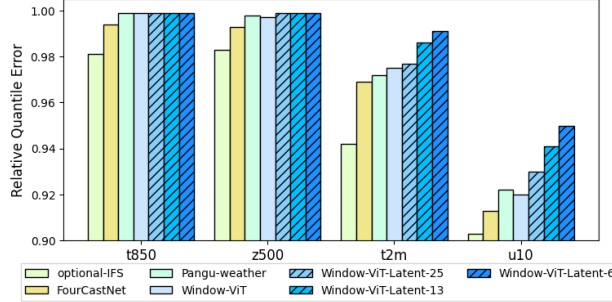

*Figure 7.* Extreme weather forecasting performance on the RQE metric. For easier visual comparison in the figure, the RQE is transformed using the formula $1 + 10 \cdot$ RQE, with the optimal performance being closer to 1.

**Superior preservation of extreme events.** As illustrated in Fig.7 and Fig.6 (a), WTL achieves superior performance in extreme weather forecasting compared to the pixel-space baseline. The SEDI scores confirm that WTL maintains high accuracy in detecting extreme occurrences. Furthermore, the RQE scores demonstrate that WTL effectively minimizes errors in extreme value magnitudes. By operating in the latent space, WTL achieves superior accuracy and sharpness in extreme weather forecasting compared to WT and other baselines like Pangu-Weather and FourCast-Net, confirming the advantage of latent space prediction for preserving extreme values.

**Competitive overall forecast skill.** Fig.6 (b) shows that WTL remains competitive with WT in terms of overall fore-

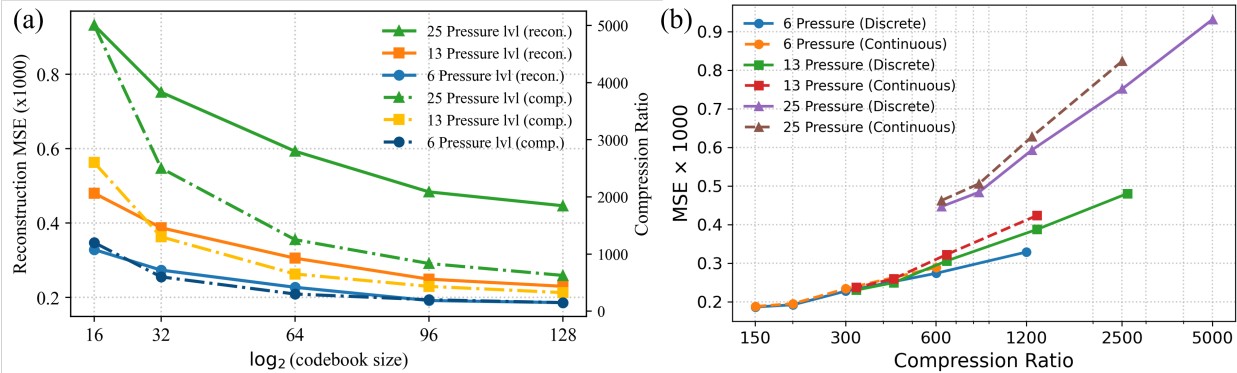

*Figure 8.* Ablation studies evaluated on the atmospheric temperature variable. (a) Ablation study on compression ratio and reconstruction quality of the WLA under varying input pressure levels (6, 13, 25 layers) and codebook sizes ($2^{16}$ to $2^{128}$). (b) Ablation study on the BQM, which can be seen as an comparsion of discrete and continuous latent space.

cast skill. Both models exhibit performance comparable to Pangu-Weather, and surpass baselines like Operational IFS and FourCastNet. Notably, WTL consistently produces sharp and accurate forecasts while adapting to multiple PVS inputs. These results suggest that conducting weather forecasting in the latent space is an efficient strategy: it matches the performance of pixel-space models while substantially reducing data storage and computational costs.

#### 4.5.2. EVALUATION ON PRECIPITATION FORECASTING

To further substantiate the utility of our proposed framework beyond general weather forecasting , we conduct evaluations on the precipitation forecasting task. Precipitation forecasting presents a unique challenge due to its probability distribution, which typically peaks strongly at zero and exhibits a heavy tail towards positive values (Kurth et al., 2023). Following the post-processing methodology in (Kurth et al., 2023; Zhou et al., 2022), we construct a lightweight mapping network to project the 6-hour forecast states of the weather prediction model onto 6-hour accumulated precipitation.

Following the experimental protocol in (Kurth et al., 2023), we employ the RQE metric and ERA5 dataset to assess the model. We compare two variants of our framework: the Window-ViT-Precipitation (WTP) in the pixel space, and the Window-ViT-Latent-Precipitation (WTLP) in the latent space.

The experimental results in Figure S1 shows that the performance of the pixel-based WTP is comparable to that of FourCastNet (Kurth et al., 2023), and the latent-based WTLP consistently outperforms both WTP and FourCast-Net. It demonstrates a stronger capability in characterizing the heavy-tailed distribution of precipitation data in the latent space. While WTLP shows improvements over other baselines, its performance remains lower than that

of Operational-IFS. This highlights that accurate extreme precipitation forecasting remains an challenge for purely data-driven models and warrants further investigation.

### 4.6. Ablation Study

**Pressure Levels and Codebook Size**. To identify the optimal balance between compression efficiency and reconstruction quality, we conducted ablation studies on the atmospheric temperature variable using the upper-air dataset. We evaluated the WLA across three input configurations (6, 13, and 25 pressure levels) and five codebook sizes ($2^{16}, 2^{32}, 2^{64}, 2^{96}, 2^{128}$).

The general ablation results, illustrated in Fig. 8(a), demonstrate the inherent trade-off in our framework: the compression ratio is inversely proportional to the codebook size but positively correlated with the number of input pressure levels (consistent with Eq. 1). Conversely, reconstruction quality improves with larger codebook sizes but decreases as the input data dimensionality grows. Based on these observations, we selected a codebook size of $2^{128}$ for our final model, as further increasing the size yields diminishing returns in reconstruction quality.

**BQM Discretization**. To further investigate the role of discretization by BQM in data compression and reconstruction, we compared our discrete framework against a continuous baseline. To ensure a fair comparison under identical data compression rates, we constructed a continuous variant where the BQM is replaced by a linear layer, which maps features to a latent space representation stored as a float32 with $log_2(codebook\ size)//32$ channels. The channel dimensions were adjusted such that the total bit-width remains constant. For example, for a codebook size of $2^{128}$, 128 binary channels are compared against 4 float32 channels.

As shown in Fig.8(b), the discrete latent space achieves reconstruction performance comparable to the continuous

baseline at lower compression ratios. Notably, at higher compression ratios, the discrete model significantly outperforms its continuous counterpart. This suggests that the BQM effectively preserves essential semantic information during discretization, offering superior efficiency over a bitrate-matched continuous representation.

**Impact of Adversarial Loss**. To address the concern of whether the improved sharpness benefits from the loss function rather than from the proposed framework, we conducted an ablation study comparing our approach against a pixel-space weather forecasting model trained with adversarial loss. Details can be found in Section A.5. The results validate the effect of proposed framework, which effectively decouples the objectives: the WLA utilizes GAN loss to ensure sharp reconstruction, while the latent-space weather forecasting model employs Binary Cross-Entropy loss to stably learn temporal dynamic features in the latent space.

## 5. Conclusion

We presented the Weather Latent Autoencoder, a novel method for learning efficient latent representations of weather data. WLA solves key issues of pixel-space approaches, including prediction smoothness and inaccuracy, single PVS limitations, and prohibitive costs. Our resulting ERA5-Latent dataset compresses ERA5 data from 244.34 TB to 0.43 TB. WLA and the ERA5-Latent dataset offer a robust foundation for advancing meteorological research within latent space. Future work will target improved reconstruction and higher-resolution weather datasets.

## Acknowledgements

This work was supported in part by the National Natural Science Foundation of China (Grant No. 42522112); the Natural Science Foundation of Jiangsu Province (Grant No. BK20250065); the AI and AI for Science Project of Nanjing University (Grant No. 020914380141); the Youth Innovation Team of China Meteorological Administration (Grant No. CMA2024QN02); and the Joint Fund for Meteorology of the National Natural Science Foundation of China (Grant No. U2442217). This work was supported in part by the Shanghai Artificial Intelligence Laboratory.

## Impact Statement

The development and training phases of the presented models necessitate substantial computational resources, inherently leading to significant energy consumption. This energy expenditure constitutes a critical environmental concern, contributing materially to the carbon footprint and other associated ecological impacts. Recognizing these externalities, we emphasize the importance of mitigating strategies focused on energy sourcing. Specifically, transitioning towards renewable and low-carbon energy infrastructure for powering computational tasks is paramount to lessening the environmental burden associated with large-scale model training.

Adopting sustainable energy solutions can demonstrably reduce the ecological ramifications of the computational pipeline, aligning technological advancement with environmental stewardship. It is incumbent upon the research community to proactively evaluate and address the environmental costs inherent in deploying computationally demanding methodologies. Key mitigation approaches encompass not only the adoption of sustainable energy but also advancements in energy-efficient hardware architectures and the continuous pursuit of algorithmic optimization to reduce computational overhead. Promoting such holistic sustainable computational practices is crucial for ensuring that progress in artificial intelligence does not inadvertently exacerbate environmental challenges, but rather contributes responsibly to future development.

### 5.1. Reproducibility statement

To facilitate reproducibility, we provide comprehensive training details for the weather latent autoencoder in the supplementary materials. Code, ERA5-latent data, and pretrained models are available at WLA.

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

# A. Supplementary Material

## A.1. Experimental Details of Weather Latent Autoencoder

### A.1.1. ERA5 DATASET DETAILS

The ERA5 dataset (Hersbach et al., 2020) is a global atmospheric reanalysis product from the European Center for Medium-Range Weather Forecasts (ECMWF) and serves as a standard for evaluating the Weather Latent Autoencoder in comparison to other models. This dataset is highly valued in climate research due to its high spatial resolution of 0.25° and extensive weather coverage. The dataset is temporally partitioned into training sets (1979–2021, 233.48 TB), validation sets (2022, 5.43 TB), and test sets (2023, 5.43 TB). To meet various weather-related needs, we organized three categories of variables within the ERA5 dataset: upper-air, surface, and precipitation variables.

For **upper-air variables**, we selected three configurations spanning 25, 13, and 6 pressure levels, each containing six core weather variables: geopotential height ($z$), longitudinal wind speed ($u$), meridional wind speed ($v$), vertical velocity ($w$), atmospheric temperature ($t$), and specific humidity ($q$), in which variables are presented by abbreviating their short name and pressure levels (e.g., q1000 denotes the specific humidity at a pressure level of 1000 hPa).

The **surface variables** include two subsets: an 8-variable set comprising 10m v-component of wind (10v), 10m u-component of wind (10u), 100m v-component of wind (100v), 100m u-component of wind (100u), 2m temperature (t2m), Total cloud cover (tcc), surface pressure (sp) and Mean sea-level pressure (msl); and a streamlined 4-variable subset (10v, 10u, tcc, msl).

The **precipitation variables** cover cumulative hourly precipitation over six intervals (tp1h, tp2h, tp3h, tp4h, tp5h, tp6h), with two additional single-variable subsets (tp1h and tp6h).

### A.1.2. IMPLEMENTATION DETAILS

In meteorological downstream tasks, the selection of the six basic upper-air variables is generally fixed, while the pressure levels selected vary significantly depending on the task. Conversely, for surface data, the specific variables selected vary. To adapt to the data diversity and accommodate the distinct physical properties of atmospheric data, we employ a variable-specific training strategy. Specifically, we train separate WLAs for each of the six upper-air variables to capture multi-pressure-level dependencies, and distinct models for the surface and precipitation variables to exploit intra-variable correlations. This design treats different variables as independent modalities (Chen et al., 2023a), ensuring that the unique physical laws governing each variable do not interfere during feature extraction, thereby maximizing reconstruction fidelity.

All WLAs are trained with identical configurations across weather variables. The models are optimized for 500K steps on 4 Tesla A100 GPUs. The codebook size is set to $2^{128}$ for upper-air and surface variables, while precipitation variables employ a reduced codebook size of $2^{32}$ due to their higher compressibility. The input data are processed through patches of size $15 \times 14$ with a stride of $10 \times 10$ and a padding of $2 \times 2$.

Following the architectural insights of (Hansen-Estruch et al., 2025), where decoder upscaling demonstrated significant reconstruction benefits without comparable encoder gains, we design the VAEformer with asymmetric depths: a 16-layer encoder versus a 32-layer decoder. Training employs the AdamW (Loshchilov & Hutter, 2019) optimizer with an initial learning rate of $3.2 \times 10^{-5}$, batch size 8, and a hybrid learning schedule that combines a linear warm-up phase increasing the learning rate from $3.2 \times 10^{-6}$ to $3.2 \times 10^{-5}$, followed by a cosine decay phase.

Following the settings in BSQ (Zhao et al., 2024), the total loss function comprises three parts: entropy loss, MSE loss, and GAN loss. Among them, entropy loss is used to improve the utilization rate of the codebook, and MSE loss and GAN loss are used to improve the accuracy and clarity of the reconstruction results.

### A.1.3. BASELINE

The compared state-of-the-art compression methods include Elic (He et al., 2022), IEN (Xie et al., 2021), VQVAE (van den Oord et al., 2017), VQGAN (Esser et al., 2021), and VAEformer (Han et al., 2024a). ELIC (He et al., 2022) introduces an efficient architecture that utilizes unevenly grouped space-channel contextual adaptive coding, striking a balance between rate-distortion performance and computational complexity. IEN (Xie et al., 2021) enhances the compression pipeline by proposing a more powerful invertible encoding network, which improves the modeling of the latent representations' distribution. The comparison also includes methods based on discrete representations. VQ-VAE (van den Oord et al., 2017) is a foundational model that learns a discrete codebook for latent variables, effectively preventing posterior collapse.

Building upon this, VQGAN (Esser et al., 2021) combines the discrete quantization of VQ-VAE with the high-fidelity synthesis power of Transformers and GANs. Lastly, VAEformer (Han et al., 2024a) adapts a variational transformer architecture specifically for the extreme compression of large-scale scientific data, leveraging the transformer's ability to capture complex long-range dependencies.

For the Elic (He et al., 2022), IEN (Xie et al., 2021), we use the code in CompressAI 2 (Ballé et al., 2017) to reimplement and retrain them. For the VQVAE (van den Oord et al., 2017), VQGAN (Esser et al., 2021), we fine-tune their pre-trained models on meteorological data. For VAEformer (Han et al., 2024a), since its pre-trained model uses data consistent with this study, we directly use its pre-trained model for comparison.

The most competitive baseline among all baselines is VAEFormer. VAEFormer is specifically designed for meteorological data compression. It uses the atmospheric circulation transformer block as a basic block to effectively capture the characteristics of atmospheric circulation. Meanwhile, VAEFormer includes two stages: pre-training and fine-tuning. The pre-training stage trains a VAE-style transformer encoder that generates the compressed latent representation, and a transformer-based decoder restores it to the reconstructed data. The fine-tuning stage trains another encoder and decoder to predict the mean and scale hyperpriors for the Arithmetic Encoder and Decoder process, which further losslessly compresses the data via entropy coding.

## A.2. Compression Comparison Details

Due to significant differences in numerical ranges among weather variables, we included a "Variable Std" row as a reference. Generally, variables with higher variances exhibit larger reconstruction errors. Since the number of input variables in WLA influences both the compression ratio and reconstruction quality, we evaluated its performance using the maximum input configurations: 25 pressure levels for upper-air variables, 8 variables for surface variables, and 6 variables for precipitation variables. The compression ratio and bpsp values reported for WLA in Table 1 correspond to its performance on these three variable categories, respectively.

## A.3. Experimental Details of Downstream

### A.3.1. ERA5-LATENT DATASET DETAILS

The partitioning of ERA5-Latent data and the selection of PVS remain consistent with Section 4.1. By utilizing the high compression rate of WLA, the ERA5-Latent dataset reduces the original 244.34 TB of data down to 0.43 TB, while providing a unified representation for multiple PVS. These subsets include three for upper-air variables corresponding to 25, 13, and 6 pressure levels, two for surface variables (4 and 8 variables), and three for precipitation variables (tp1h, tp6h, tp1-6h). To facilitate the computation of pixel metrics, the ERA5-Latent dataset also incorporates the raw pixel data for July 2023, which has been compressed using the Lempel-Ziv-Markov chain-Algorithm (LZMA) and occupies 0.117 TB of storage. LZMA is a widely used lossless compression algorithm developed by Igor Pavlov and implemented via the Python standard library.

Building upon the ERA5-Latent dataset, deep learning models for large-scale meteorological research can utilize the unified latent representation to seamlessly handle multiple PVS, making the models adaptable for a wide range of meteorological tasks and scenarios. Moreover, processes that typically require large datasets, such as training, validation, and testing, can be conducted using the compact latent data, significantly reducing both storage and data computational costs. For tasks requiring only a limited amount of pixel data, pixel data from a single month within the ERA5-Latent dataset can be used to compute metrics and visually compare model outputs with the original data.

### A.3.2. MODEL DETAILS

Window-ViT employs the multimodal encoder-decoder from FengWu (Chen et al., 2023a) and the backbone of the weather latent autoencoder, conducting weather forecasting task in the pixel space. It treats each upper-air variable and the grouped surface variables as distinct modalities, and uses local and global self-attention to model complex atmospheric dynamics. In contrast, Window-ViT-Latent omits the downsampling and upsampling components present in WT while keeping all other components unchanged, conducting weather forecasting task in the latent space.

### A.3.3. VARIABLE AND METRIC DETAILS

The variables used in the weather forecasting experiment are consistent with the settings of Pangu-Weather, including five upper-air variables (z, u, v, t, q) and four surface variables (10v, 10u, t2m, msl).

The SEDI metric classifies each pixel into extreme or normal weather using high quantile thresholds (90%, 95%, 98%, and 99%) and then calculates the hit rate, a value closer to 1 indicates a more accurate prediction of extreme weather.

The RQE (Kurth et al., 2023) is an indicator used to assess a model's ability to capture extreme values within a given field, such as wind speed or precipitation. Its calculation involves summing up the relative difference between the predicted and true values across a range of high quantiles (like the 90th to 99.99th percentiles), focusing on the most extreme events. Essentially, the RQE reveals systematic biases: a negative RQE suggests the model is consistently under-predicting the magnitude of these extremes, while a positive RQE would indicate over-prediction.

### A.3.4. COMPARED MODELS

The Operational Integrated Forecasting System of the European Centre for Medium-Range Weather Forecasts (ECMWF) is widely regarded as the world's leading global numerical weather prediction (NWP) system (Bougeault et al., 2010). It utilizes a comprehensive Earth system model and an advanced data assimilation system to produce forecasts for the medium-range and beyond. The IFS simulates the complex interactions within the Earth's atmosphere and its coupled systems, solving mathematical equations that govern their dynamics and physics to predict future weather conditions.

Pangu-Weather is a deep learning-based weather forecasting system (Bi et al., 2023). It is a data-driven model that employs a 3D deep neural network architecture to capture intricate patterns in atmospheric data. Trained on decades of global reanalysis data, Pangu-Weather demonstrates strong performance in medium-range forecasting for various atmospheric variables. Unlike traditional NWP models, it does not explicitly solve physical equations but learns the evolution of weather patterns directly from historical data.

## A.4. Downstream Task: Precipitation Forecasting

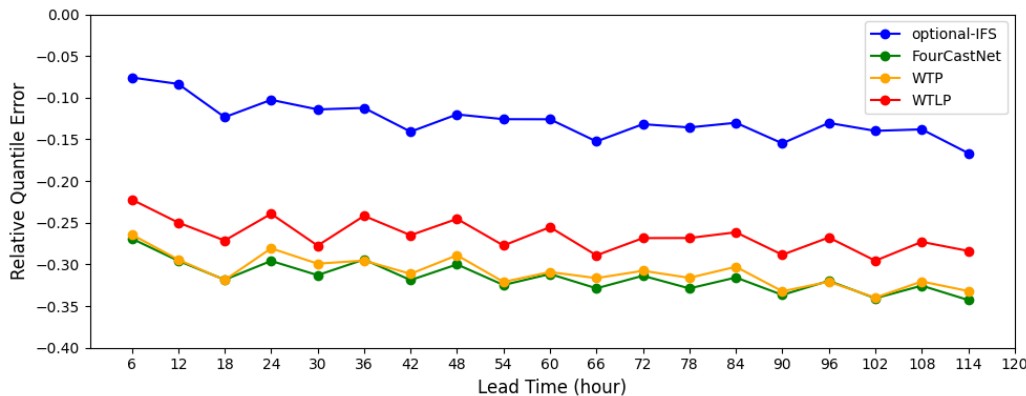

*Figure S1.* Comparison on the precipitation forecasting task. The smaller the RQE is than 0, the worse the performance in precipitation forecasting.

## A.5. Impact of Adversarial Loss

To address the concern regarding whether the improved sharpness in extreme event prediction is merely a methodological artifact of the loss function rather than a benefit of our proposed latent framework, we conducted an ablation study comparing our approach against a pixel-space baseline trained with adversarial loss.

We trained the baseline pixel-space WT with an additional Generative Adversarial Network (GAN) loss, referred to as Window-ViT-GAN (WTG). All other hyperparameters and training configurations remained identical to the standard WT baseline to ensure a fair comparison.

The results in Figure S2 show that the WTG model failed to surpass the standard baselines. Instead of improving extreme

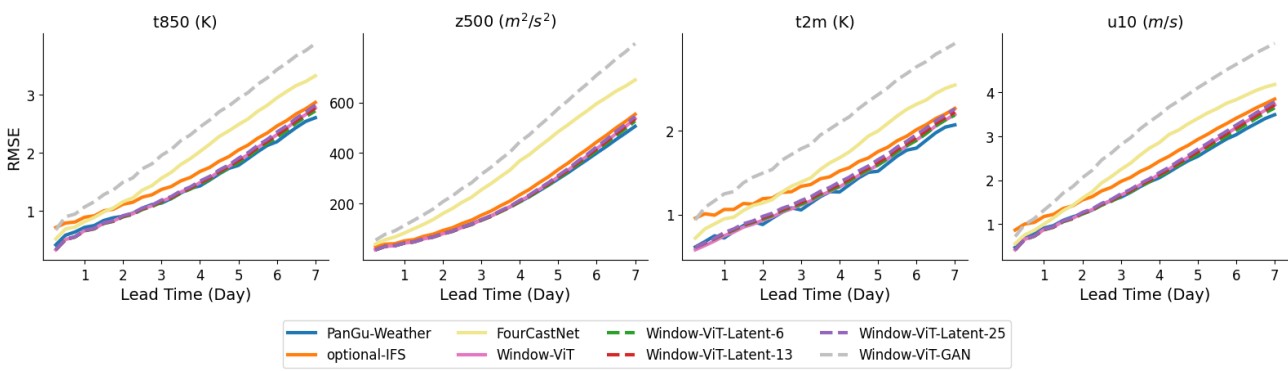

*Figure S2.* Comparison on the weather forecasting task over a 7-day period. The number following "Window-ViT-latent" indicates the number of input pressure levels.

event prediction, the instability compromised the model's ability to learn correct temporal dynamics, resulting in inferior predictive performance across standard metrics compared to both the vanilla WT and other baselines. These results underscore the non-triviality of applying adversarial training to global weather forecasting. Merely altering the loss function in pixel space leads to optimization difficulties. This validates the design of our Latent Framework, which effectively decouples the objectives: the WLA utilizes GAN loss to ensure sharp reconstruction, while the WTL employs Binary Cross-Entropy loss to stably learn temporal dynamics in the latent space.

### A.6. Limitations and Expectations

Although the weather latent autoencoder effectively transforms weather data from pixel space to latent space, we acknowledge certain limitations, particularly regarding the computational trade-offs and scalability.

First, to accommodate the distinct physical properties of atmospheric variables and ensure high-fidelity reconstruction, we trained separate autoencoders for distinct variable groups (upper-air, surface, and precipitation). This approach is both data-intensive and computationally expensive. Consequently, both the model training and the generation of the comprehensive ERA5-Latent dataset entail a substantial upfront computational investment. We frame this as a necessary trade-off: a significant one-time cost is incurred to enable substantial long-term reductions in storage and computational overhead for downstream applications and the broader research community. Furthermore, during the deployment phase of WLA, the unavoidable processes of data encoding and decoding will still introduce additional computational overhead.

Second, the current architecture is primarily optimized for ERA5 data at a resolution of $0.25°$. Scaling to higher-resolution datasets (e.g., HRES at $0.09°$) remains a challenge, as the exceedingly large global weather images impose significant computational demands during training.

In future work, we plan to build upon the foundation of the ERA5-based autoencoder and employ efficient fine-tuning techniques to adapt the model to both global and regional datasets at higher spatial resolutions. This strategy aims to further mitigate computational barriers and facilitate a paradigm shift towards conducting scalable weather research within the latent space.

### A.7. Visualization of Reconstructed Results

To provide a detailed assessment of our model's reconstruction fidelity and to transparently analyze potential information loss due to compression, this section presents a comprehensive visual analysis. We examine six key variables from the ERA5 dataset: Atmospheric temperature at 850hpa (T850), Geopotential height at 500hpa (Z500), Temperature at 2 meters (t2m), U-component of wind at 10 meters (10u), Mean sea-level pressure (msl), and 6-hour accumulated precipitation (tp6h).

As illustrated in Figures S1 through S6, we present three case studies for each variable. Each case includes a side-by-side comparison of: (a) the original field from ERA5, (b) the field reconstructed from our model's latent representation, and (c) a difference map (i.e., reconstruction error) to precisely identify the magnitude and location of any information loss.

The reconstruction and difference maps demonstrate that our model achieves superior reconstruction quality across this

diverse set of atmospheric variables. The difference maps reveals that some localized information loss occur. This loss is not random but is systematically correlated with regions of extreme values in the original data. The spatial patterns of this information loss are variable-dependent, as the distribution of extreme values is intrinsically tied to the physical nature of each atmospheric field. For example:

1. For 6-hour accumulated precipitation (tp6h), information loss predominantly manifests in localized cores of intense convective rainfall, such as within tropical cyclones or severe thunderstorms.

2. For temperature at 2 meters (t2m), larger reconstruction errors are more likely to appear in areas with extreme temperatures, such as the polar regions or hot deserts.

3. For geopotential height at 500hpa (Z500), the most notable discrepancies are found at the centers of deep, low-pressure troughs or high-pressure ridges, which represent the maxima and minima of the atmospheric wave patterns.

While having some information loss, the model proves highly effective at preserving the fine-grained, small-scale extreme values that are vital for meteorological applications. For instance, the visualizations confirm the retention of sharp gradients in Z500 fields associated with atmospheric troughs and ridges, the intricate structures of precipitation bands (tp6h) within storm systems, and the tight pressure contours (msl) that define the core of cyclones. This robust performance underscores the model's ability to effectively capture and represent the essential features of the original data.

This detailed visual analysis provides a transparent and nuanced understanding of our model's performance, confirming its high fidelity while also characterizing the predictable and physically grounded nature of the minor information loss.

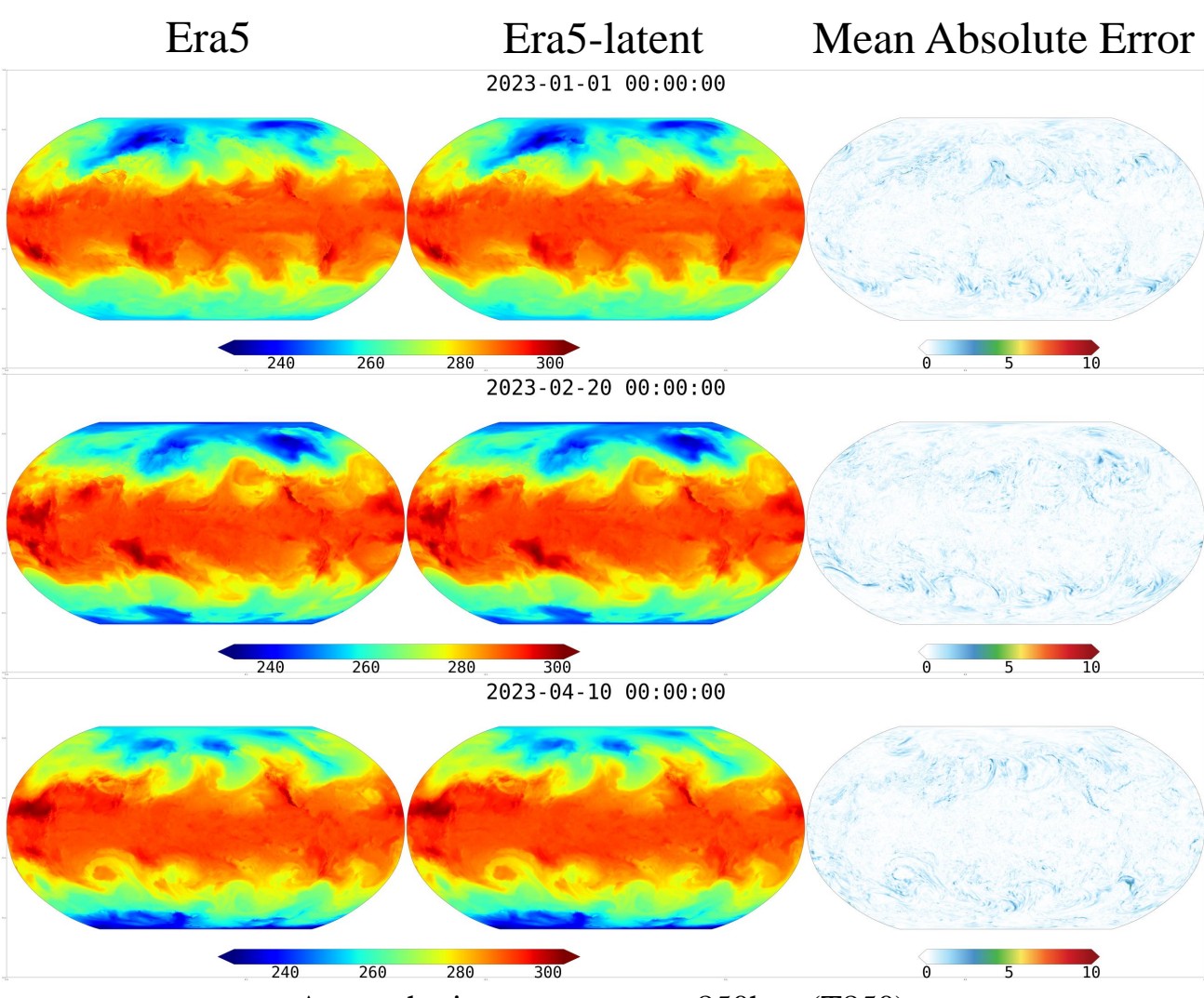

Atmospheric temperature at 850hpa (T850)

*Figure S3.* Visualization samples of T850 on the ERA5 and the compressed ERA5-Latent. From the left to the right column: ERA5, ERA5-Latent, and their absolute error map.

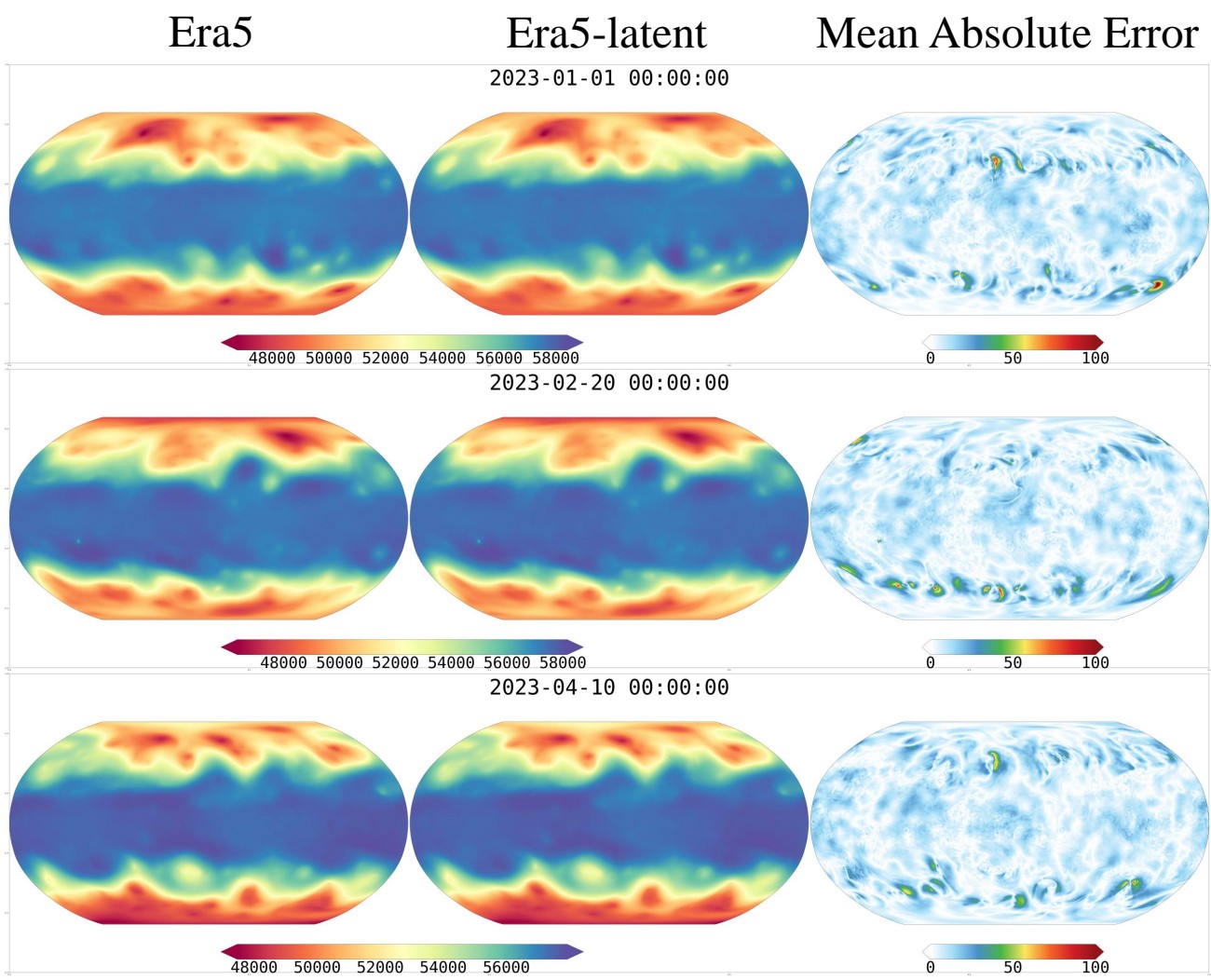

Geopotential height at 500hpa (Z500)

*Figure S4.* Visualization samples of Z500 on the ERA5 and the compressed ERA5-Latent. From the left to the right column: ERA5, ERA5-Latent, and their absolute error map.

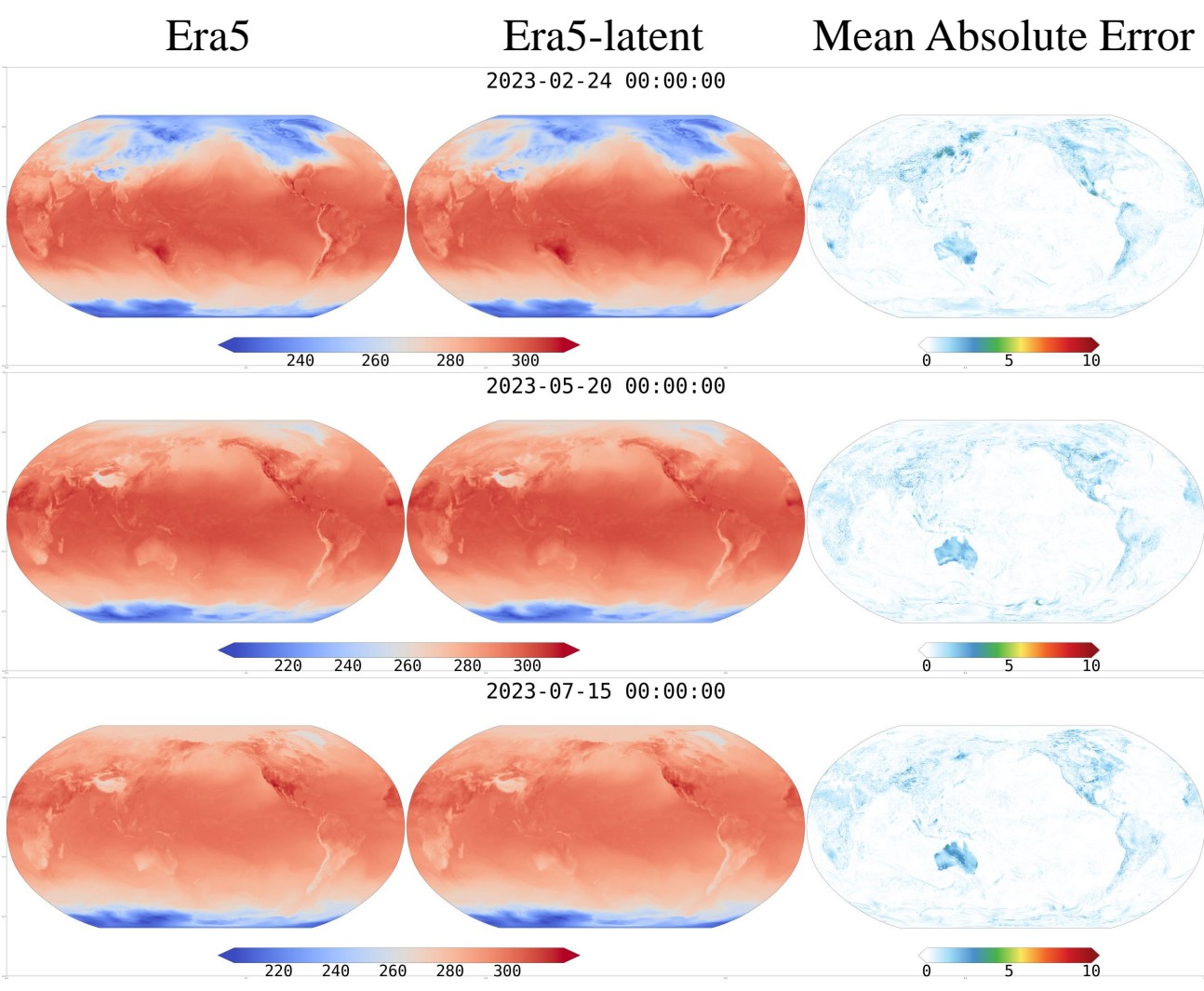

Temperature at 2 meter (t2m)

*Figure S5.* Visualization samples of t2m on the ERA5 and the compressed ERA5-Latent. From the left to the right column: ERA5, ERA5-Latent, and their absolute error map.

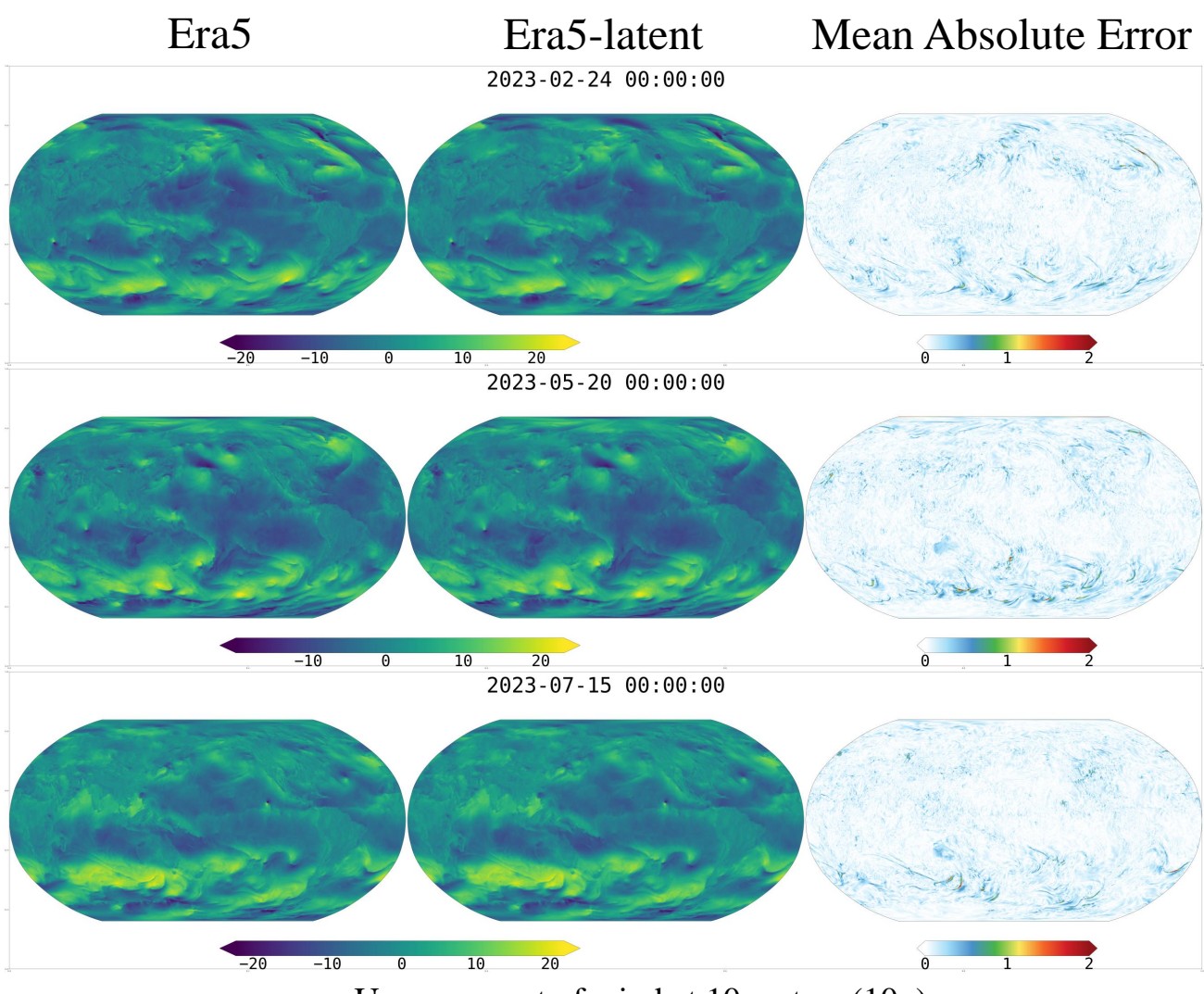

*Figure S6.* Visualization samples of 10u on the ERA5 and the compressed ERA5-Latent. From the left to the right column: ERA5, ERA5-Latent, and their absolute error map.

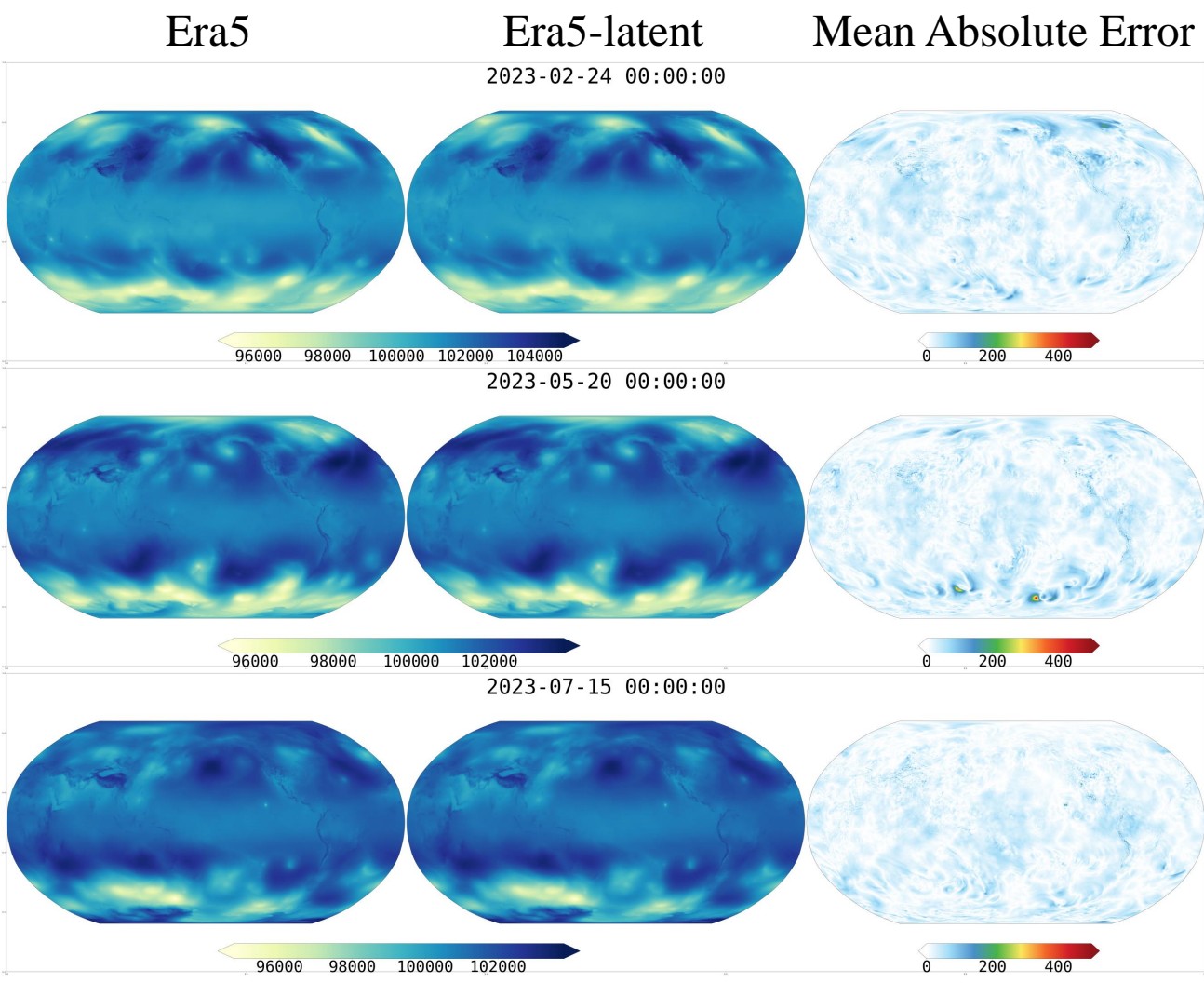

*Figure S7.* Visualization samples of msl on the ERA5 and the compressed ERA5-Latent. From the left to the right column: ERA5, ERA5-Latent, and their absolute error map.

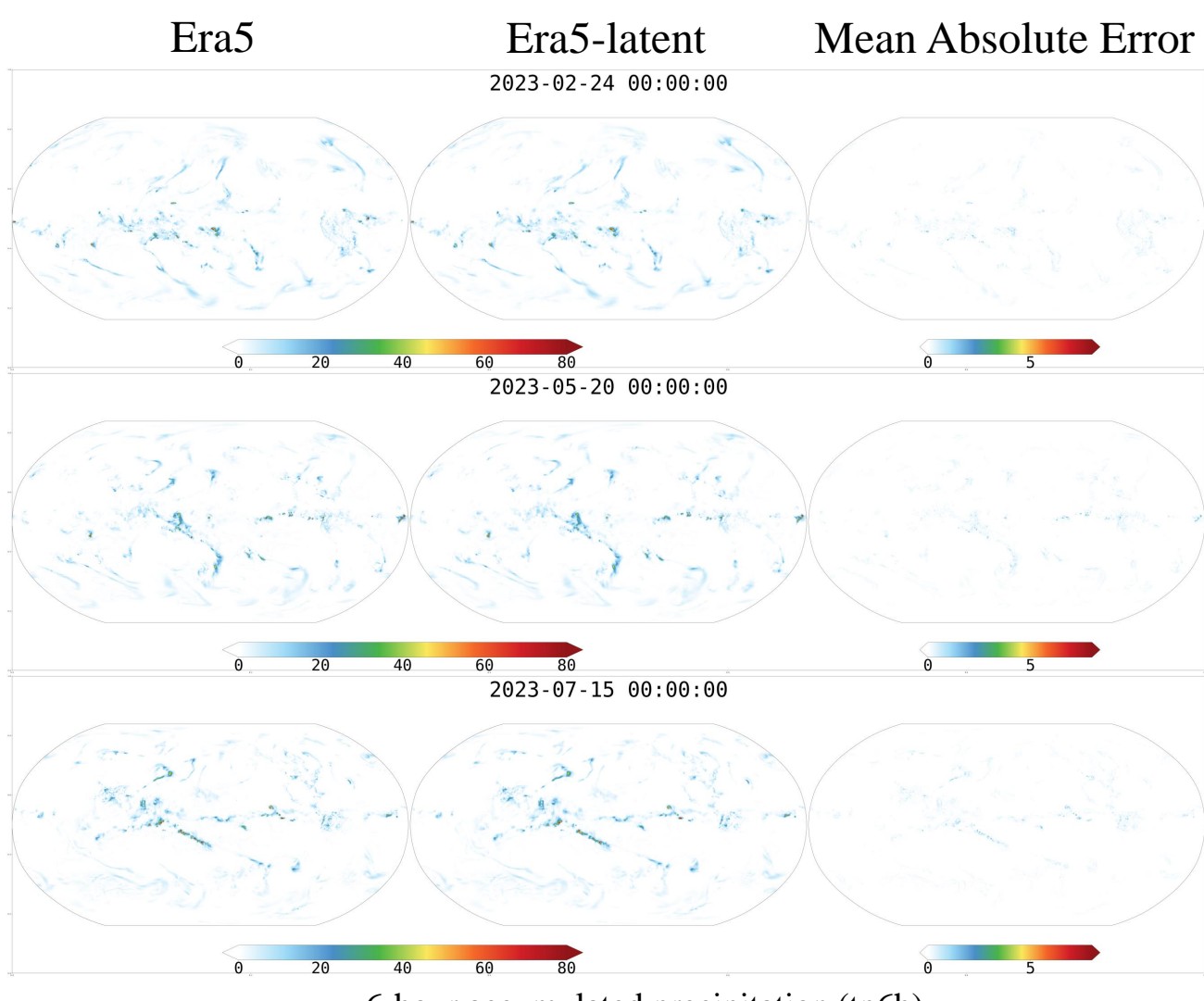

6-hour accumulated precipitation (tp6h)

*Figure S8*. Visualization samples of tp6h on the ERA5 and the compressed ERA5-Latent. From the left to the right column: ERA5, ERA5-Latent, and their absolute error map.

