# OpenReview forum: "Transforming Weather Data from Pixel to Latent Space"
_ICML.cc/2026/Conference — ICML 2026 spotlight_

### Official Review · Reviewer_NAJ3 · 2026-03-07

**Soundness:** 2
**Presentation:** 2
**Significance:** 3
**Originality:** 3
**Overall Recommendation:** 4
**Confidence:** 4

**Summary:**

The paper proposes a weather-specific autoencoder (Weather Latent Autoencoder, or WLA) that projects atmospheric data into a compressed latent space, allowing downstream tasks to be modelled at reduced computational and storage costs. A key component of the WLA is the pressure-variable unified module, which leverages hypernetworks to enable generalisation to pressure levels (and potentially variables) unseen during training. The authors demonstrate the efficacy of their approach through two main sets of experiments primarily using ERA5 data: 1) evaluating the autoencoder's reconstruction ability both in-distribution and out-of-distribution across varying pressure levels and data sources, and 2) assessing performance on downstream forecasting tasks, comparing latent-space modelling (enabled by WLA) against pixel-space baselines. Additionally, the authors release a compressed version of the widely used ERA5 dataset, achieving a >500x compression ratio.

**Compliance With Llm Reviewing Policy:**

Affirmed.

**Final Justification:**

I am maintaining my score of Weak Accept. The paper makes a significant and original contribution, particularly by introducing hypernetworks to generalise to unseen pressure levels—an approach I really liked. During the rebuttal, the authors successfully resolved my core technical concerns by clarifying computational costs, adjusting their claims, and providing the necessary frequency-space analysis, all of which will improve the paper's soundness. However, I chose not to raise my score because the empirical evaluation still lacks comparisons to what I would consider truly SOTA weather models. Furthermore, the manuscript requires structural and presentation refinements to clearly deliver its main messages. While the authors have committed to implementing these changes, I remain somewhat cautious about the extent to which the overall clarity of the work will ultimately be improved.

**Key Questions For Authors:**

Q1. Do the authors have any insights about how effective hypernetworks would be in the extrapolation, rather than interpolation regime? I.e., train on up to 800hPa, and test on >850hPa. I would imagine this to be a harder generalisation task, but any empirical insights would be useful.
Q2. Could the authors provide a comprehensive frequency space analysis that shines light on how the reconstructions between latent and pixel space compare spectrally?
Q3. Could the authors provide a clear computational cost analysis between the latent and pixel-space modelling? This would ideally include a mention about storage cost for the multiple WLA models, as well as training cost (cost of WLA training + cost of latent-space training versus cost of pixel-space training), and inference-time cost. Something more than “500k steps on 4 Tesla A100 GPUs” would be useful for contextualisation.
Q4. Could the authors address (some of) the points about WLA Compression (W2.1)?
Q5. Do the authors have any insights about whether the OOD experiment would hold equally for other variables, or whether some qualitative differences might emerge because different variables have different dynamics?
Q6. ERA5 and HRES are both reanalysis. Have the authors ever tried the proposed approach using observational data?
Q7. Figure 6a seems to indicate that using fewer pressure levels for Window-ViT leads to better SEDI metrics (although the colour scheme makes the results slightly hard to interpret). Could the authors comment on that? How come using fewer pressure levels leads to better results on this downstream task? And what are the differences in RMSE, it’s very hard to differentiate between the different configurations based on the plots in Figure 6b.

**Limitations:**

Yes, although some clarification about how computationally expensive the technique is would be useful.

**Strengths And Weaknesses:**

**Strengths**

S1. **Hypernetworks for generalisation** -  I found this to be a clean solution for generalising to unseen pressure levels. In meta-learning it’s had various degrees of success depending on the difficulty of the task/generalisation, but I do believe it is a good idea in this setting, especially in the interpolation regime (i.e. train on 500hPA and 600hPa, and query at 550hPa). This is clearly demonstrated on the temperature variables by the out-of-domain ERA5 experiment.

S2. **Good Ablations** -  The section on ablation studies (Section 4.4) provides a useful apples-to-apples comparison for the most effective compression strategy (continuous versus discrete), as well as useful insights about how the performance varies with codebook size.

S3. **Addressing downstream tasks** - The integration of the WLA in experiments on downstream tasks allows for a more comprehensive evaluation of the technique not only as a compression model, but also in a more practical setting — where the strengths of latent-space modelling can be leveraged.

S4. **Relevance to the community** - Lately there have been multiple attempts to perform latent-space modelling for weather prediction, but I do not think there has so far been convincing evidence that latent-space models can be competitive with pixel-space ones. Related to this, it is interesting to note that the recent work [1] that compares multiple probabilistic methods in a downsampled latent space, construct this latent space through bilinear interpolation on the encoder side, and only train the decoder, highlighting the difficulty of training an efficient encoder-decoder model for weather data.

**Weaknesses**

W1. **Unsupported Claims and Cost Analysis**

- W1.1. **Sharper results, dealing with smooth outputs** - One of the main weaknesses of the papers is the lack of concrete evidence regarding the shaper results and incomplete discussion about weather models producing smooth outputs.
  - For the authors to claim that the model is producing sharper results, I would expect some frequency-space analysis that supports it. I do not think the visualisations in A.7. are enough to support this claim (and they do not show the pixel-space reconstructions to support the “sharper results than pixel-space models” claim).
  - Claiming that weather models produce smooth results is more nuanced than presented in the paper. Depending on the modelling framework, models do this to different extents: deterministic models suffer more from oversmoothing, but probabilistically-trained ones tend to show better spectral matches. This should be discussed in the paper.

- W1.2. **Alternatives are not suitable for contemporary deep learning applications** (L114 left) - The authors claim that the compressed ERA5 version provided by  Weatherbench is not suitable for applications, but I think the jury is still out there. A lot of the evidence points towards the fact that the majority of the skill achieved by these models is derived from the low spatial frequencies anyway (because that’s where the majority of the energy of the spectrum is concentrated). So at least by the metrics we are measuring (e.g., RMSE), the models do not lose much performance by operating on the lower-resolution data, meaning that they might actually be suitable for contemporary applications (at least by the metrics we are looking at). An example of this is ArchesWeather [2] which, despite working at a 1.5$^\circ$ resolution, still achieves competitive performance with SOTA higher-resolution models. I do not think there is consensus in the field about what the right resolution to operate at is (and indeed this is complicated by a shift towards observation-based models where they no longer operate at a fixed resolution - Aardvark [3]), so I would be more careful about this claim. Moreover, to show that WLA provides a more efficient compression of the ERA5 dataset as opposed to the Weatherbench data, the authors should provide a comparison to the best model trained on Weatherbench data, which is not the case.

- W1.3. **Computational cost** - The authors claim that “data storage and computational costs are significantly reduced as model training, validation, and inference primarily utilize low-storage latent features, restricting pixel-space operations to final metric evaluation phases.” L181 right. This is true in a development scenario, but as soon as the model goes into deployment, the decoding would likely need to happen often (depending on the downstream task). This is compounded by the use of separate WLA models for different groups of variables (eight in total if I am not wrong: 6 for upper-variables, 1 for surface, and 1 for precipitation).

W2. **Empirical Evaluation and Baselines**

- W2.1. **WLA Compression Metrics**:  Overall this section convinces me that WLA is likely better than any other baselines that compress weather data, but it’s unclear how much information loss the WLA leads to. The weighted RMSE on its own is not enough, and the analysis would be much stronger if it included frequency space analysis, and potentially a comparison to the weighted RMSE achieved by a SOTA model at one-step.
  - Reporting the results on a select set of variables is incomplete. The authors could provide a per-variable average (i.e., averaged over pressure levels for upper-air ones, averaged over all surface variables, averaged over all times for precip).
  - Having a VQGAN/VQVAE at the same compression ratio would provide a more conclusive comparison between the methods, it might just be that the compression ratio used is too aggressive.
  - To get a better grasp of the amount of information lost could the authors compare the weighted RMSE of the WLA with the weighted RMSE at the smallest lead time (6h) of a SOTA pixel-space model on some key variables? My rationale is the following: The one-step differences are fairly predictable and SOTA models show good performance there. If WLA has a greater reconstruction loss than these differences, it imposes a fundamental restriction about how a latent-space model that leverages WLA could compare to a SOTA pixel-space model. Alternatively, maybe the authors could provide a comparison to a 1h prediction of a strong NWP model.

- W2.2. **Out-of-distribution (OOD) Generalisation**: The experiment is performed on temperature only. To me, it is not entirely clear that the generalisation will hold for the other variables because it might depend on the dynamics of each variable (how quickly they vary with altitude).

- W2.3. **Downstream Task Comparison**:
  - **Ambiguous phrasing** - “This experiment almost takes the same configuration as the state-of-the-art model [4]” - L364 right. First, this is not very formally expressed, and needs clarification about why it is only “almost” the same configuration.
  - **Unclear computational comparisons to baselines** - The authors provide little detail that allow for a clear, interpretable comparison of the training cost of each approach. Besides the one-off cost of the 8 WLA models, do they allocate the same compute budget (and how is that defined) to Window-ViT and Window-ViT-Latent? Is it based on the same number of optimisation steps? How does that translate into wall clock time?
  - **Gap to SOTA weather model** - Pangu is a good model, but it is well known that since Pangu appeared, other models that achieve better skill have emerged. It would be good to show a comparison to the truly SOTA model out there (e.g., FGN [5]) just to understand what the gap to operational ML models is.

- W3. **Structure and Clarity**
  - **Results subsections** - Overall I found the structure of the results section unclear. A better way to structure it would be to have a subsection on 1) Compression quality and another one on 2) Integration into downstream tasks, each then having subsequent subsubsections.
  - **Figures and formatting** - Figure 7 is referenced in the text before Figure 6, which negatively impacts the readability of the manuscript. I also had trouble understanding what recon. and comp. mean in Figure 7a). Finally, the labels are generally too small, but completely unreadable in Figure 8.
  - **Mathematical formalism** - The training of WLA involves three different loss terms — for completeness, I think it would be better to mathematically define each of them.

W4. **Minor**
- “The downstream task further demonstrates that task models can apply to multiple PVS with low data costs in latent space and achieve superior performance compared to models in pixel space.” - Abstract. This is not grammatically correct, and I am unsure what the authors mean here.
- PanGu versus Pangu inconsistencies
- Optional instead of Operational throughout figures, in L729 Appendix A.3.4, etc.
- Interpretation of figure S1 in A.4 (or context) is lacking.
- L245 right - “High-resolution pixel space is reserved only for data sparse tasks, such as metric calculations and result visualizations.” - The usage of “data sparse tasks” is unclear here - I guess the authors meant that the decoding to pixel space only needs to be performed at evaluation stage, but they should rephrase.
- Typos: L255 right “is serves”
- L669 - “The most challenging baseline” - I am assuming the authors meant competitive?
- L356 left - “improved sharpness is benefit from loss function” - incorrect phrasing.
- WTL is first mentioned in L363 (left) without being defined up until that point of the manuscript.
- Window-ViT is first mentioned in L383 (right) but there is no reference. Based on my understanding from A.3.2., this is based on the FengWu [4] architecture, but this should be clarified in the main. I would also prefer to see an architecture description in A.3.2, that would help better contextualise the model without requiring to look at the FengWu paper.
- The authors sometimes mention six upper-air variables (A.1.1.), sometimes mention five (excluding vertical velocity). Are the compression results on six variables, and the downstream tasks one on five variables? Could the authors clarify?

**References**

1. Kossaifi, J., Kovachki, N., Mardani, M., Leibovici, D., Ravuri, S., Shokar, I., … & Kautz, J. (2026). Demystifying Data-Driven Probabilistic Medium-Range Weather Forecasting.

2. Couairon, G., Singh, R., Charantonis, A., Lessig, C., & Monteleoni, C. (2024). Archesweather & archesweathergen: a deterministic and generative model for efficient ml weather forecasting.

3. Vaughan, A., Markou, S., Tebbutt, W., et al. (2024). "Aardvark Weather: end-to-end data-driven weather forecasting".

4. Chen, K., Han, T., Gong, J., Bai, L., Ling, F., Luo, J. J., ... & Ouyang, W. (2023). Fengwu: Pushing the skillful global medium-range weather forecast beyond 10 days lead.

5. Alet, F., Price, I., El-Kadi, A., Masters, D., Markou, S., Andersson, T. R., ... & Battaglia, P. (2025). Skillful joint probabilistic weather forecasting from marginals.

---

> ### Author Rebuttal · Authors · 2026-03-31
>
> Dear Reviewer NAJ3:
>
> We sincerely thank the reviewer for their constructive feedback. We have categorized comments into overarching concerns (C), mapped them to weaknesses (W), questions (Q) and limitations (L), and provided our answers (A) below.
>
> > **C1. Frequency domain analysis of prediction results (W1.1, W2.1, Q2)**
>
> **A1.** We selected the u10 variable and the radially averaged power spectrum metric for frequency domain analysis under the settings in Section 4.6. As shown in the [link](https://anonymous.4open.science/r/conf_paper_image-5089/frequency%20analysis.png), the latent space model (Window-ViT-Latent) exhibits higher energy in the high-wavenumber region, demonstrating that it generates sharper outputs. Furthermore, results on the SEDI and RQE metrics in Section 4.6 also confirm this.
>
> > **C2. Discussion on probabilistic models (W1.1)**
>
> **A2.** We have added content in the Related Work section regarding how probabilistic models can help mitigate output smoothness.
>
> > **C3. Inappropriate phrasing regarding WeatherBench (W1.2)**
>
> **A3.** We agree that our phrasing regarding WeatherBench in L114 was inappropriate, and we acknowledge its immense value in many applications. Our intention was to highlight that in scenarios requiring high-resolution data—such as capturing small-scale extreme values—un-downsampled high-resolution data is essential. Our goal is not to replace WeatherBench, but to provide an efficient avenue to access high-resolution ERA5 data. We have revised L114, removing the "restricting their suitability" claim, and instead clarified the necessity and high cost of using high-resolution ERA5 data.
>
> > **C4. Computational cost of WLA in deployment scenarios (W1.3, L)**
>
> **A4.** In the deployment phase, both WLA and other existing compression models inevitably incur the encoding and decoding computational cost. We have added a discussion clarifying this cost in Appendix A.6.
>
> > **C5. One-step prediction comparison between WLA and SOTA models (W2.1, Q4)**
>
> **A5.** We have added the one-step prediction RMSE of Operational-IFS, FourCastNet, and Pangu to Table 1. The results show that WLA's reconstruction outperforms these baselines, confirming that WLA's reconstruction quality is robust and does not impose a restriction on the performance of latent space models.
>
> > **C6. Displaying reconstruction performance across all variables (W2.1, Q4)**
>
> **A6.** Because there are significant differences in the means and variances across various variables, providing the average loss can be misleading. Therefore, we have added a new section in the Appendix detailing the reconstruction loss for all variables to illustrate the model's reconstruction capabilities.
>
> > **C7. Comparison with VQVAE/VQGAN at the same compression ratio (W2.1, Q4)**
>
> **A7.** Please refer to A4 in our response to Reviewer uBcT.
>
> > **C8. Generalization experiments limited to the temperature variable (W2.2, Q5)**
>
> **A8.** Please refer to A9 in our response to Reviewer DQVR.
>
> > **C9. Ambiguous phrasing in downstream task settings (W2.3, Q3)**
>
> **A9.** We have revised the phrasing in L363 to clarify the similarities and differences in the experimental setup.
>
> > **C10. Need for detailed computational cost comparisons (W2.3, Q3)**
>
> **A10.** Please refer to A4 in our response to Reviewer DQVR.
>
> > **C11. Comparison with the latest SOTA weather prediction models (W2.3)**
>
> **A11.** Because several of the latest SOTA models (e.g., FGN) have not open-sourced code or weights, conducting a comparison is currently challenging. We will prioritize adding these comparisons in our future work as resources become available.
>
> > **C12. Optimization of structure, clarity, figures, and formatting (W3, W4)**
>
> **A12.** We sincerely appreciate your meticulous reading. We have carefully corrected and refined the content as you suggested.
>
> > **C13. Effectiveness of hypernetworks in pressure level extrapolation (Q1)**
>
> **A13.** Extrapolation is indeed more challenging. In practice, we mitigate this by constructing training data to encompass the maximum and minimum pressure levels, thereby transforming potential extrapolation scenarios into more reliable interpolation tasks.
>
> > **C14. Transferring the method to observational data (Q6)**
>
> **A14.** We have not yet applied this approach to observational data. Compared to reanalysis data, observational data exhibits strong spatial sparsity and multi-modal characteristics, making direct application of existing compression models difficult.
>
> > **C15. Further clarification regarding Figure 6 (Q7)**
>
> **A15.** Regarding Figure 6a, we believe you are referring to Window-ViT-Latent. The improved SEDI performance is likely because the compression model achieves better reconstruction under fewer pressure levels. For Figure 6b, Window-ViT-Latent-6 has a slightly lower RMSE overall, likely because predicting fewer variables reduces the error and the results can be reconstructed better.

---

> > ### Author Rebuttal · Reviewer_NAJ3 · 2026-04-01
> >
> > I thank the authors for their detailed rebuttal. In general, my concerns have been adequately addressed by the authors' agreeing to make the necessary modifications. As stated before, I believe this work has clear value, but it could benefit from presentation refinement. While I appreciate the authors incorporating the majority of the feedback, I remain somewhat cautious about whether the overall clarity of the work will be improved. For this reason, I am maintaining my current score (which remains on the positive side anyway).
> >
> > Moving forward, I strongly encourage the authors to: expand the frequency domain analysis to better substantiate the "sharper outputs" claim; explicitly state all necessary experimental details; and include a comparison to SOTA baselines as resources become available (a comparison to Aurora [1] should already be feasible).
> >
> > [1] Bodnar, C., Bruinsma, W. P., Lucic, A., Stanley, M., Allen, A., Brandstetter, J., ... & Perdikaris, P. (2025). A foundation model for the Earth system. Nature, 641(8065), 1180-1187.

---

> > > ### Author Response · Authors · 2026-04-02
> > >
> > > Dear Reviewer NAJ3,
> > >
> > > We sincerely thank you for reviewing our rebuttal, acknowledging that your concerns have been fully resolved, and maintaining your positive evaluation of our work.
> > >
> > > We greatly appreciate your constructive suggestions. We take your feedback on presentation refinement seriously and will ensure that necessary analysis are involved and experimental details are explicitly stated, and the writing is polished for clarity.

---

### Official Review · Reviewer_s1oc · 2026-03-11

**Soundness:** 3
**Presentation:** 3
**Significance:** 3
**Originality:** 3
**Overall Recommendation:** 5
**Confidence:** 4

**Summary:**

The authors propose a weather latent autoencoder (WLA) that generates rich embeddings from weather data in pixel space. It aims to decouple weather reconstruction from downstream tasks to better learn generalist representations and store weather information as embeddings, reducing data storage and computational costs.

The WLA includes a pressure-variable unified module (PVUM) to transform pressure-variable subsets (PVS) into a unified latent space, a variational autoencoder, and a binary quantization module (BQM) to compact the latent space.

The authors generated the ERA5-Latent dataset with embeddings from multiple PVSs to encourage the community to work on latent representations, reducing data storage and computational overheads.

Experiments were conducted to evaluate reconstruction quality, compression ratio, and bits per sub-pixel on ERA5, showing that the method achieves a favorable trade-off between metrics compared to competing methods. Results on out-of-domain ERA5 samples and ablation studies demonstrate the method's generalization capacity and support the design choices of codebook size, BQM discretization, and loss function.

**Compliance With Llm Reviewing Policy:**

Affirmed.

**Final Justification:**

The authors successfully answered my questions, I would thus recommend this work toward acceptance.

**Key Questions For Authors:**

### Key Questions For Authors
1. L.124: "Our framework avoids this by directly using latent space data, reducing data calculation costs." To what extent can your framework work directly on the latent space with new weather data, given that it still needs to generate embeddings with the encoder?
2. How does your method compare to existing weather foundation models that generate low-dimensional embeddings [1, 2, 3]? Are these methods comparable in both compression and downstream applications?
3. What are the benefits and impacts of using hypernets instead of concatenating metadata tokens with $X$ and processing them all with a single linear layer?
4. Would it be meaningful to perform the codebook size ablation study with respect to downstream task performance rather than the pretext task (reconstruction)? It would be interesting to observe whether a smaller codebook size maintains downstream task performance.
5. What is the exact architecture of the VAEformer used? Have you tried different backbone sizes to analyze potential impacts on embedding quality?
6. Do the RMSE scores correspond to the reconstruction error of the input data? Section 4.2 and the caption of Table 1 could be improved to clarify this detail.
7. Since reconstructing input data is not the only way to compare methods, how would embeddings generated by competing methods in Table 1 perform on the presented downstream tasks? Similarly, how would weather foundation models [1, 2, 3] perform on these downstream tasks using only generated embeddings?
8. One concern about leveraging embedding-based datasets (i.e., ERA5-Latent) for downstream applications is that important physical properties or processes from the raw data may be lost. How would one measure the latent space, or constrain it by design, to ensure physical properties are maintained?

### Additional Comments
1. Figure 2: Consider mentioning examples of metadata types instead of integer numbers to make the figure clearer.
2. Figure 3: Be more specific about which components constitute the hypernet.
3. Figure 8: Please increase text size.
4. Ensure color consistency between methods in Figure 6(b) with 6(a) and 8 for better readability.

### References:

[1] J. Schmude et al., Prithvi WxC: Foundation Model for Weather and Climate. In ArXiv 2024.

[2] C. Bodnar et al., A foundation model for the Earth system. In Nature 2025.

[3] I. Price et al., Probabilistic weather forecasting with machine learning. In 2024.

**Limitations:**

Yes

**Strengths And Weaknesses:**

### Strengths:
1. The application field is important for tackling the growing amount of weather data and forecasting weather extremes that evolve due to climate change.
2. The contributions are well motivated by clearly stating the limitations of current work on pixel-level weather data with single pressure-variable subsets (PVS), and how the proposed work addresses them.
3. The overall WLA framework, including the variational autoencoder, pressure-variable unified module (PVUM), binary quantization module (BQM), and latent space framework (LSP), is well explained.
4. Exhaustive experiments including weather variable reconstruction, out-of-domain generalization, weather and precipitation forecasting demonstrate the method's relevance, with design choices supported by comprehensive ablation studies.
5. The ERA5-Latent dataset is provided to enable development of methods based on generated embeddings, allowing focus on lightweight representations rather than large tensors of raw recordings.

### Weaknesses:
1. Insufficient related work and comparison with existing weather foundation models [1, 2, 3], which could, in practice, generate low-dimensional latent spaces similar to those proposed by the authors.
2. Lack of downstream task experiments comparing the quality of embeddings generated by competing methods against the proposed method (see Questions section).
3. Additional ablation studies could be performed on the impact of HyperNets and the VAEformer encoder architecture (see Questions section).


### References:

[1] J. Schmude et al., Prithvi WxC: Foundation Model for Weather and Climate. In ArXiv 2024.

[2] C. Bodnar et al., A foundation model for the Earth system. In Nature 2025.

[3] I. Price et al., Probabilistic weather forecasting with machine learning. In 2024.

---

> ### Author Rebuttal · Authors · 2026-03-31
>
> Dear Reviewer s1oc:
>
> We sincerely thank the reviewer for their constructive feedback and insightful questions. We have summarized your concerns as **C** (mapping to Weaknesses **W**, Questions **Q**, and Additional Comments **Ad.C**) and provided our detailed answers as **A** below.
>
> > **C1. Insufficient introduction of weather foundation models in related work (W1)**
>
> **A1.** We have expanded the related work section to include recent advancements in weather foundation models.
>
> > **C2. To what extent can the framework operate directly in the latent space? (Q1)**
>
> **A2.** Please refer to A3 of our response to Reviewer uBcT.
>
> > **C3. Comparison of embeddings between weather compression models and foundation models (Q2, Q7)**
>
> **A3.** Direct comparison between WLA and weather foundation models is difficult due to their differing objectives: foundation models prioritize downstream task embeddings, while WLA focuses on unified representation and high compression. Consequently, foundation models excel in task performance, while WLA leads in data compression. For instance, the lightest Aurora achieves a compression ratio of ~10.25 under the settings in Section 4.2, significantly lower than WLA.
>
> Crucially, these approaches are complementary. Foundation models can be built within WLA’s latent space (similar to VQGAN-based pretraining), allowing them to adapt to diverse variable subsets without the separate linear layers required by Aurora.
>
> > **C4. Advantages of using HyperNets over a fixed linear layer (W3, Q3)**
>
> **A4.** HyperNets offer two key advantages over a fixed linear layer:
>
> 1. **Adaptive Data Mapping:** Unlike fixed layers that cannot handle variable-length features, HyperNets use a Transformer to generate dynamic parameters, enabling adaptive mapping for varying data.
>
> 2. **Metadata Correlation Mining:** HyperNets uncover underlying correlations to boost out-of-domain generalization. (See response A3 to Reviewer uBcT for a detailed PVUM comparison.)
>
>
> > **C5. Impact of different codebook sizes on downstream tasks (Q4)**
>
> **A5.** Generally, an autoencoder's reconstruction performance establishes the upper bound for prediction performance within its latent space. Therefore, a smaller codebook reduces reconstruction quality, which inherently degrades downstream task performance. To validate this, we evaluated forecasting performance on t2m using autoencoders with codebook sizes of 2³², 2⁶⁴, and 2¹²⁸ unter the settings in Section 4.6. The results shown below confirm that models operating in a latent space with a smaller codebook exhibit lower downstream performance.
>
> | Codebook Size | SEDI-90% | SEDI-95% | SEDI-98% | SEDI-99% | RQE |
> | --- | --- | --- | --- | --- | --- |
> | $2^{32}$ | 0.90 | 0.88 | 0.87 | 0.86 | 0.941 |
> | $2^{64}$ | 0.92 | 0.91 | 0.91 | 0.90 | 0.965 |
> | $2^{128}$ | 0.93 | 0.92 | 0.91 | 0.91 | 0.977 |
>
> > **C6. VAEformer architecture and the impact of backbone size on embedding quality (Q5)**
>
> **A6.** VAEformer adapts the ViT architecture by replacing standard patch embeddings with our **PVUM module** and employing a **hybrid local-global attention** mechanism (inspired by CRA5's ACT [1]). Following standard practice to balance performance and efficiency, we selected a single, optimal backbone size based on previous studies. We adopted the CRA5 [1] backbone size as a baseline. Following ViTok[2], we specifically **expanded the decoder** to ensure high-fidelity weather data reconstruction.
>
> > **C7. Downstream task performance of embeddings from competing compression methods (W2, Q7)**
>
> **A7.** While other compression methods generate embeddings, their sole objective is data compression. They do not consider direct utilization of the data in the latent space, meaning their latent spaces lack the regularization required for downstream applications. In contrast, BQM simultaneously binarizes and uniformly maps latent features onto a spherical space, establishing a well-regularized latent space explicitly designed to support downstream tasks.
>
> > **C8. Impact of data embeddings on physical properties (Q8)**
>
> **A8.** We discuss the potential impact of data embeddings on physical properties in Appendix A.7. Our analysis indicates that while there is some information loss during compression, crucial physical properties are preserved. In future work, we plan to incorporate physics-informed loss functions to further constrain the latent space.
>
> > **C9. Clarifications on RMSE and figure/text optimizations (Q6, Ad.C 1-4)**
>
> **A9.** The RMSE scores correspond to the reconstruction error of the input data. We appreciate your helpful suggestions for improving clarity, and we have revised mentioned text, tables and figures.
>
> [1] T. Han et al., CRA5: Extreme Compression of ERA5 for Portable Global Climate and Weather Research via an Efficient Variational Transformer. In arxiv 2024.
>
> [2] P. Hansen-Estruch et al., Learnings from Scaling Visual Tokenizers for Reconstruction and Generation. in arxiv 2025.

---

> > ### Author Rebuttal · Reviewer_s1oc · 2026-04-03
> >
> > I thank the authors for considering my review in details.
> > It answers my questions and hopefully will help to improve the quality of the submission.
> > I would recommend toward accepting this high-quality work, and thus will increase my score.

---

> > > ### Author Response · Authors · 2026-04-03
> > >
> > > Dear s1oc,
> > >
> > > We sincerely thank you for your positive evaluation and for recognizing the quality of our work. We are glad to hear that our responses have fully addressed your concerns. We will ensure that the discussions and clarifications from the rebuttal are carefully incorporated into the final version to further enhance the paper.

---

### Official Review · Reviewer_DQVR · 2026-03-12

**Soundness:** 3
**Presentation:** 3
**Significance:** 3
**Originality:** 3
**Overall Recommendation:** 4
**Confidence:** 4

**Summary:**

The paper identifies three limitations of performing weather-related tasks (forecasting, downscaling) in pixel space—smooth model outputs due to task uncertainty affecting reconstruction of small-scale extremes, limited applicability to a single pressure-variable subset (PVS) while different tasks need different PVS compositions, and high storage and computational costs (ERA5 on the order of hundreds of TB). It proposes the Weather Latent Autoencoder (WLA), which transforms weather data from pixel space to a unified latent space. WLA combines (1) a Pressure-Variable Unified Module (PVUM) that maps any PVS via metadata and a hypernetwork to a unified feature space, (2) a VAEformer encoder-decoder backbone (from CRA5) for compression and reconstruction, and (3) a Binary Quantization Module (BQM) with spherical normalization and sign-based binary quantization for compact bitwise tokens. Reconstruction is decoupled from downstream tasks so that task models operate in latent space, improving sharpness and reducing data cost. The authors introduce ERA5-Latent, compressing ERA5 from 244.34 TB to 0.43 TB (about 566×). Experiments compare compression and reconstruction (Table 1) against ELIC, IEN, VQVAE, VQGAN, and VAEformer on weighted RMSE, compression ratio, and bits per sub-pixel (bpsp), and compare downstream forecasting in pixel space (Window-ViT, WT) versus latent space (WTL) and precipitation models (WTP vs WTLP). WLA achieves strong reconstruction on many variables and downstream latent-space models match or slightly exceed pixel-space baselines on extreme-event metrics (SEDI, RQE) while remaining competitive on overall RMSE. Ablations cover codebook size, number of input pressure levels, and discrete versus continuous latent representation.

**Compliance With Llm Reviewing Policy:**

Affirmed.

**Final Justification:**

I maintain my Weak Accept recommendation. The paper presents a technically solid and practically relevant latent-space framework for weather data compression and downstream modeling, and the rebuttal adequately addressed my main concerns by clarifying the compression metrics, strengthening the evidence for PVUM, extending the generalization discussion beyond temperature, and adding more detail on variable statistics, computational cost, release details, and limitations. While some evaluation scope and presentation issues still remain, I believe the work makes a useful contribution that the weather and climate ML community can build on.

**Key Questions For Authors:**

1. How is bpsp computed for WLA versus the baselines in Table 1? If WLA does not use the same entropy coding as VQVAE/VQGAN, could you clarify so that the compression ratio and bpsp comparison is interpretable?

2. The main text states that WLA has “higher compression ratios, lower bpsp” versus existing methods; in Table 1, VQVAE and VQGAN report compression ratio 1100 and bpsp 0.029, versus WLA 625.9 and 0.051. Could you reconcile this (e.g. clarify which “existing methods” are meant or add a sentence acknowledging that some baselines achieve higher compression)?

3. Is there an ablation or analysis that isolates the contribution of PVUM (e.g. versus a fixed linear mapping or per-PVS adapter) to reconstruction quality or downstream performance?

4. Figure 5 reports WLA on temperature across pressure levels and on HRES. Why was temperature chosen for this generalization experiment, and is there a comparison of other compression methods (ELIC, IEN, VQVAE, VQGAN, VAEformer) on the same out-of-domain or per-pressure-level setup?

5. For the forecasting evaluation (Fig. 6, Fig. 8), do you have or could you add a brief discussion of the distribution of predictor and target variables (e.g. variance, tail heaviness) and of variable difficulty? This would help readers interpret why performance differs across variables (e.g. t850, z500, t2m vs u10) and whether strong performance on some variables reflects method strength or easier predictors.

**Limitations:**

The paper includes an Impact Statement but no dedicated Limitations section. It does not discuss dependence on ERA5 and chosen PVS configurations, the lack of predictor/target distribution discussion for the forecasting evaluation, the fact that Fig. 5 generalizes only on temperature with no comparison to other methods in that setting, the restriction of pixel-space evaluation to July 2023, training time and resource use, or potential risks of GAN-based reconstruction (e.g. mode collapse or artifacts). A short limitations paragraph would strengthen the paper. Otherwise, the above may be used as suggestions.

**Strengths And Weaknesses:**

**Soundness**
On the strengths side, the three limitations and the link to PVUM, decoupling, and latent-space operation are clearly set up. PVUM (metadata-driven hypernetwork, adaptive mapping) and BQM (spherical norm, binary quantization) are described; the VAEformer backbone is attributed to CRA5. Table 1 reports weighted RMSE as reconstruction error (reconstructed versus original field), alongside compression ratio and bpsp, against several baselines. Ablations are reported on compression strength (codebook size, number of pressure levels) and on discrete (BQM) versus bitrate-matched continuous latent (Section 4.4, Fig. 7). Downstream comparison of pixel-space versus latent-space task models (WT vs WTL, WTP vs WTLP) is presented; WTL is reported to achieve better extreme-event metrics (SEDI, RQE) while remaining competitive with WT on overall RMSE, and the paper states that conducting forecasting in latent space matches pixel-space performance while reducing storage and compute.
On the weaknesses side, the paper does not give variable-specific reasons for WLA’s superiority on particular variables (e.g. w500, w700, TCC, SP, TP6H) beyond general mechanisms (PVUM, BQM, modality-specific architectures, GAN loss). It does not discuss the practical value of comparing methods when RMSE is already very low (e.g. Q700, Q1000). The main text claims WLA has “higher compression ratios, lower bpsp” versus “existing methods”; in Table 1, VQVAE and VQGAN achieve higher compression ratio (1100) and lower bpsp (0.029) than WLA (625.9, 0.051). The paper explains the trade-off (codebook size vs compression) and that WLA prioritizes reconstruction quality, but it does not explicitly acknowledge that VQVAE/VQGAN achieve better compression ratio and bpsp, so the narrative can overstate. Figure 5 evaluates generalization (unseen pressure levels, HRES) only for temperature; the paper does not justify why temperature was chosen or discuss variable difficulty, and it does not report other compression methods on the same out-of-domain or per-pressure-level setup. For the forecasting evaluation (Fig. 6, Fig. 8), the paper does not discuss the distribution of predictor or target variables (e.g. variance, tail heaviness) or why some variables (e.g. u10) are harder; good performance on t850, z500, t2m could reflect easier predictors rather than method strength. Pixel-space evaluation is limited to July 2023 (Appendix). Training time and resource use are not quantified. The appendix notes code and data availability but the main text does not confirm release details.

**Presentation**
The structure (limitations, method, experiments) and the alignment of three limitations with three solution directions are clear. Figures 1–4 illustrate pixel vs latent, WLA architecture, PVUM, and the Latent Space Framework. Table 1 and Figs 6, 7, 8 support the narrative. Terminology is mostly consistent; the acronym PVUM and the role of BQM are introduced. The use of “codebook size” for the discrete representation may invite confusion with classical VQ codebooks. Linking the Latent Space Framework (Fig. 4) more explicitly to the downstream experiments would help.

**Significance**
Weather modeling and extreme-event prediction are important for climate and applications. Reducing storage and compute and enabling flexible use of multiple PVS in latent space is practically valuable. ERA5-Latent could lower the barrier for large-scale weather ML. The finding that task models can operate in latent space with comparable or slightly better accuracy (especially on extremes) and lower data cost is useful. The scope is weather and climate ML; generalization beyond the reported setup (e.g. other variables, other reanalyses) is not fully demonstrated, and the lack of predictor/target distribution discussion and of quantification of training cost limits how much impact can be assessed.

**Originality**
The combination of PVUM (unified representation for arbitrary PVS), VAEformer, and BQM in a single weather latent framework, plus the decoupled reconstruction-from-task design and ERA5-Latent, is a clear contribution. The backbone is adapted from CRA5; the novelty lies in the weather-specific architecture (PVUM, BQM), the latent-space workflow, and the scale of the compressed dataset. Hypernetworks for modality adaptation and binary quantization exist elsewhere; the originality is in their integration for multi-PVS weather data.

---

> ### Author Rebuttal · Authors · 2026-03-31
>
> Dear Reviewer DQVR:
>
> We sincerely thank the reviewer for their detailed and constructive feedback. We have categorized the reviewer's concerns (**C**) and mapped them to the respective Weaknesses (**W**), Questions (**Q**) and Limitations (**L**). Soundness, Presentation, Significance, and Originality in Weakness are denoted as **W.So**, **W.P**, **W.Si**, and **W.O**, respectively. Our corresponding answers (**A**) are detailed below.
>
> > **C1. Rationale for variable selection and discussion of model superiority based on variable characteristics (W.So, Q4)**
>
> **A1.** We have expanded the experimental section to discuss the model's superior performance in the context of variable characteristics. Specifically, we added detailed explanations regarding the distinct traits of various upper-air, surface, and precipitation variables.
>
> > **C2. Comparison when RMSE is already extremely low, e.g., Q700, Q1000 (W.So)**
>
> **A2.** Although the absolute RMSE values for Q700 and Q1000 are very small, this is primarily due to their inherently low means and variances. If variables like Q700, Q1000, w500, and w1000 are standardized by their respective means and variances, their RMSEs are actually on a comparable scale. Therefore, it is both appropriate and standard practice to use consistent metrics across all variables for an equitable comparison.
>
> > **C3. Better compression ratios of VQVAE/VQGAN (W.So, Q2)**
>
> **A3.** Please refer to Response A4 for Reviewer uBcT.
>
> > **C4. Further details on experiments and computational costs (W.So, W.Si)**
>
> **A4.** We have enriched the experimental details in the Appendix. This includes a comprehensive breakdown of the training, inference, and storage costs for the compression models, as well as for both the latent-space and pixel-space models in downstream tasks. We have also clarified the rationale behind our dataset setup and provided specific details regarding the open-source release of our code and data.
>
> > **C5. Calculation of BPSP in Table 1 and interpretability of the comparison (W.So, Q1)**
>
> **A5.** Both Bits Per Sub-Pixel (BPSP) and Compression Ratio (CR) are calculated directly from the raw and compressed meteorological data, making them independent of the specific encoding method used. The conversion formula is BPSP = BPP / CR, where BPP (Bits Per Pixel) is typically 32 for meteorological data. Because WLA and the baseline models use the exact same calculation method for these metrics, the comparisons are highly interpretable.
>
> > **C6. Independent contribution of PVUM (Q3)**
>
> **A6.** Please refer to Response A3 for Reviewer uBcT.
>
> > **C7. Comparison with other compression methods in generalization experiments (Q4)**
>
> **A7.** Baseline models utilize fixed linear layers for data encoding, which confines them to a single PVS. Consequently, these baseline models lack the architectural capacity to perform this specific generalization experiment.
>
> > **C8. Distribution and difficulty of variables in downstream forecasting tasks (Q5)**
>
> **A8.** We have added a statistical overview of the variables involved in the downstream tasks, including their mean, variance, and tail heaviness. We also included a brief discussion on the relative prediction difficulty of different variables. This addition will help readers better interpret why performance varies across variables.
>
> > **C9. Model generalization to other variables (W.Si)**
>
> **A9.** Our model seamlessly adapts to unseen pressure levels and out-of-domain data by leveraging the hypernetwork within PVUM to thoroughly mine correlations among metadata. This mechanism is fundamentally variable-agnostic. To empirically validate this, we tested the model on additional upper-air variables (longitudinal wind speed (u) and geopotential height (z)) using the identical setup from Section 4.3. The results shown below confirm that our model consistently generalizes well to unseen pressure levels and out-of-domain data across entirely different variables.
>
> | Variable | 500 | 550* | 600 | 650* | 700 | 750* | 775* | 800 | 825* | 850 | 875* | 900 |
> | --- | --- | --- | --- | --- | --- | --- | --- | --- | --- | --- | --- | --- |
> | ERA5-u | 0.532 | 0.532 | 0.534 | 0.551 | 0.569 | 0.573 | 0.577 | 0.581 | 0.578 | 0.562 | 0.549 | 0.481 |
> | ERA5-z | 11.3 | 10.2 | 9.36 | 9.41 | 9.46 | 8.83 | 8.34 | 7.64 | 7.39 | 7.21 | 7.18 | 7.14 |
> | HRES-u | 0.542 | -   | 0.548 | -   | 0.583 | -   | -   | 0.593 | -   | 0.575 | -   | 0.496 |
> | HRES-z | 12.1 | -   | 10.7 | -   | 10.8 | -   | -   | 8.82 | -   | 8.53 | -   | 8.39 |
>
> > **C10. Expanding the "Limitations" section and improving presentation (L, W.P)**
>
> **A10.** We have expanded Section 4.6 in the Appendix to include a dedicated discussion on the model's limitations, addressing the points raised above. Furthermore, we have refined the text in the main body to more explicitly link the Latent Space Framework to the downstream experiments, improving overall clarity.

---

> > ### Author Rebuttal · Reviewer_DQVR · 2026-04-04
> >
> > Thank you for the detailed rebuttal. The response addresses my main questions by clarifying the compression metrics, providing further support for PVUM, extending the generalization discussion beyond temperature, and adding more detail on variable statistics, computational cost, release details, and limitations. These clarifications improve the paper and address the main issues I raised. I will keep my current recommendation.

---

> > > ### Author Response · Authors · 2026-04-04
> > >
> > > Dear Reviewer DQVR,
> > >
> > > We sincerely thank you for your feedback and for acknowledging that our rebuttal has addressed all your concerns. We are pleased to hear that the additional clarifications and statistics have improved the paper. We will make sure to include all these updates in the final manuscript. Thank you again for your constructive review.

---

### Official Review · Reviewer_uBcT · 2026-03-13

**Soundness:** 3
**Presentation:** 3
**Significance:** 3
**Originality:** 3
**Overall Recommendation:** 5
**Confidence:** 4

**Summary:**

This paper proposes a Weather Latent Auto encoder (WLA) for the weather research to transform weather data from pixel space to latent space, enabling efficient data representation. The proposed WLA can effectively transform any pressure-variable subset from pixel space to a unified latent space, providing excellent compression and reconstruction performance. For the evaluation, the authors propose the ERA5-Latent dataset, containing large-scale ERA5 weather data with multiple pressure-variable subsets in a smaller data storage foot print and unified latent space. The experimental results demonstrate the effectiveness of proposed method.

**Compliance With Llm Reviewing Policy:**

Affirmed.

**Final Justification:**

There are several strengths of this paper as follows:
1. The design of the overall architecutres of the proposed method is of soundness, where the latent space provides more representation capability of the learned features.
2. The well-structured nature of this paper greatly enhances its readability.
3. The paper's motivation is clearly articulated. It seeks to address the issues of smooth outputs, limited applicability to a single pressure-variable subset (PVS), and high data storage-computation costs, leading the authors to propose WLA and create the ERA5-Latent dataset.

However, the visualized features in the Rebuttal lack clear interpretability, and their significance is not adequately demonstrated. I decide to keep my recommendation as accept.

**Key Questions For Authors:**

1. Could the authors provide feature visualizations to empirically support and validate the key points in the discussion?
2. Could the authors elaborate on the logical rationale and design considerations behind the proposed encoding method?
3. What are the specific novel contributions or unique features of the proposed model that distinguish it from existing approaches?

**Limitations:**

A more explicit discussion of the proposed method's limitations relative to VQVAE and VQGAN would strengthen the paper.

**Strengths And Weaknesses:**

Strength:
1. The design of the overall architecutres of the proposed method is of soundness, where the latent space provides more representation capability of the learned features.
2. The well-structured nature of this paper greatly enhances its readability.
3. The paper's motivation is clearly articulated. It seeks to address the issues of smooth outputs, limited applicability to a single pressure-variable subset (PVS), and high data storage-computation costs, leading the authors to propose WLA and create the ERA5-Latent dataset.

Weakness:
1. The paper would benefit from the inclusion of feature visualizations to support its discussion.
2. To enhance the paper's clarity, the authors should elaborate on the logical flow underlying the design of the proposed encoding method.
3. While the model design is interesting, its originality and unique features are not clearly articulated and merit further elaboration.

---

> ### Author Rebuttal · Authors · 2026-03-31
>
> Dear Reviewer uBcT:
>
> We sincerely thank the reviewer for the constructive feedback. For clarity, we have summarized your concerns (denoted as **C**), mapped them to your specific Weaknesses (**W**), Questions (**Q**) and Limitations (**L**), and provided our corresponding answers (**A**) below.
>
> > **C1. Feature Visualizations to Support Discussion (W1, Q1)**
>
> **A1.** Taking the temperature variable as an example, we have visualized the model's quantization error across different codebook sizes (see [link](https://anonymous.4open.science/r/conf_paper_image-5089/feature%20quantization%20errors.png)). The results demonstrate that the average quantization error decreases as the codebook size increases. This aligns with the analysis in BSQ [1] and supports our ablation study finding that larger codebook sizes yield better reconstruction performance.
>
> > **C2. Logical Rationale Behind the Encoding Method Design (W2, Q2)**
>
> **A2.** The core objective of our model is to transform diverse weather data into a unified latent representation while maintaining high compression rates and minimal information loss. To achieve this, we implemented designs in both the architecture and the PVUM.
>
> To map diverse inputs into a unified representation and reconstruct them, PVUM leverages the correlation between metadata and data diversity. It employs hypernetworks at both the encoder's head and the decoder's tail to convert metadata into adaptive mapping weights. Furthermore, the overall architecture utilizes a powerful VAEFormer backbone for efficient feature extraction and introduces a BQM to convert continuous features into binary representations, ensuring a high compression ratio without sacrificing significant information.
>
> > **C3. Novel Contributions and Unique Features (W3, Q3)**
>
> **A3.** The primary novelties of our model lie in the **PVUM** and the **Latent Space Framework**.
>
> - **PVUM:** Unlike methods that construct fixed linear mapping layers for different PVS, PVUM utilizes hypernetworks to generate adaptive mapping networks based on metadata. To validate its effectiveness, we compared PVUM against fixed linear layers under the settings in Section 4.4. The comparative model replaces the PVUM in the original model with PVS-specific linear layers. Linear-25 and Linear-37 respectively refer to models trained under 25-layer and 37-layer pressure level settings. As shown below, a single model utilizing PVUM achieves performance comparable to all separate models using fixed linear layers, significantly enhancing model efficiency.
>
> | Model | 500 | 550* | 600 | 650* | 700 | 750* | 775* | 800 | 825* | 850 | 875* | 900 |
> | --- | --- | --- | --- | --- | --- | --- | --- | --- | --- | --- | --- | --- |
> | Linear-25 | 0.392 | N/A | 0.384 | N/A | 0.339 | N/A | N/A | 0.292 | N/A | 0.262 | N/A | 0.216 |
> | Linear-37 | 0.388 | 0.384 | 0.381 | 0.362 | 0.334 | 0.319 | 0.301 | 0.289 | 0.275 | 0.26 | 0.241 | 0.212 |
> | WLA | 0.395 | 0.391 | 0.387 | 0.369 | 0.341 | 0.325 | 0.308 | 0.293 | 0.278 | 0.268 | 0.245 | 0.221 |
>
> - **Latent Space Framework:** The unified and compact latent space offers distinct advantages for both data compression and downstream weather tasks.
>
>   - *For data compression:* Existing compression models require decompression before use, leading to either excessive computational overhead (online decompression) or massive storage occupation (offline decompression). Our framework allows data-intensive processes like model training to occur within the latent space, reducing both storage and computational costs.
>
>   - *For weather tasks:* The framework decouples weather reconstruction from predictive tasks, mitigating the smoothing effect caused by task uncertainty and yielding sharper results. Additionally, the unified representation enables models trained in the latent space to adapt to various PVS compositions across diverse weather scenarios.
>
>
> > **C4. Limitations Relative to VQVAE and VQGAN (L)**
>
> **A4.** WLA can increase the compression ratio by reducing the codebook size, allowing for comparison with VQVAE and VQGAN at comparable compression ratios. We add a WLA with a codebook size of $2^{64}$ in Table 1 and compare it against VQVAE and VQGAN. As shown below, WLA outperforms the baseline models at a comparable compression ratio, further validating the robustness of our proposed architecture. We added content stating that WLA with a smaller codebook size can be used in scenarios requiring high compression ratios.
>
> | Method | w500 | w700 | q700 | q1000 | TCC | SP  | tp6h | Comp. Ratio | bpsp |
> | --- | --- | --- | --- | --- | --- | --- | --- | --- | --- |
> | VQVAE | 0.382 | 0.401 | 0.00108 | 0.00113 | 0.19 | 673.32 | 1.29 | 1100.0 | 0.029 |
> | VQGAN | 0.367 | 0.371 | 0.00101 | 0.00107 | 0.18 | 652.38 | 1.20 | 1100.0 | 0.029 |
> | WLA | 0.126 | 0.168 | 0.00038 | 0.00041 | 0.087 | 328.4 | 0.64 | 1251.7 | 0.025 |
>
> [1] Y. Zhao et al., Image and Video Tokenization with Binary Spherical Quantization. In ICLR 2025.

---

> > ### Author Rebuttal · Reviewer_uBcT · 2026-04-04
> >
> > I thank the authors for their comprehensive rebuttal. They have thoroughly answered my questions regarding feature visualization, motivation and originality. However, the visualized features lack clear interpretability, and their significance is not adequately demonstrated. I decide to keep my recommendation as accept.

---

> > > ### Author Response · Authors · 2026-04-06
> > >
> > > Dear Reviewer uBcT,
> > >
> > > We sincerely thank you for reviewing our initial rebuttal, maintaining your positive "Accept" recommendation, and providing further constructive feedback. We completely understand your follow-up concern regarding the interpretability and significance of our initial feature visualizations.
> > >
> > > To address this and clearly demonstrate the significance of the latent features, we have conducted a more comprehensive visualization experiment. We expanded our analysis across multiple distinct meteorological variables (z500, q850, u10, t2m) and evaluated them under various codebook sizes ($2^{16}$, $2^{32}$, $2^{64}$, $2^{128}$).
> > >
> > > 1. Enhanced Visualization Design & Interpretability:
> > >
> > >   To make the latent features interpretable, we visualized the relationship between the model's compression bottleneck and the final output. For each variable and codebook size, we visualized four specific components:
> > >
> > >   - Original Image & Reconstructed Image: To visually assess the physical fidelity of the meteorological fields.
> > >
> > >   - Reconstruction Error: The Mean Absolute Error between the original and reconstructed images, highlighting exactly where spatial and physical details are lost.
> > >
> > >   - Quantization Error: Calculated as the absolute difference between the features before and after binary quantization in the BSQ module, averaged across the channel dimension. This provides a direct, interpretable map of how much information is discarded by the latent bottleneck at specific spatial locations.
> > >
> > > 2. Demonstrated Significance:
> > >
> > >   As the codebook size increases (from $2^{16}$ to $2^{128}$), both the latent quantization error and the reconstruction error significantly decrease across all tested meteorological variables. High quantization errors are primarily concentrated around extremes, consistent with reconstruction error patterns, with a few outliers in other regions. This supports the claims made in our ablation study regarding the impact of codebook capacity on information retention, and the relationship between codebook size and quantization error align with the analysis in BSQ [1].
> > >
> > >   The detailed visualization results for each variable can be viewed via the anonymous links below:
> > >
> > >   [z500](https://anonymous.4open.science/r/conf_paper_image-5089/Visualization_z500.png)
> > >
> > >   [q850](https://anonymous.4open.science/r/conf_paper_image-5089/Visualization_q850.png)
> > >
> > >   [u10](https://anonymous.4open.science/r/conf_paper_image-5089/Visualization_u10.png)
> > >
> > >   [t2m](https://anonymous.4open.science/r/conf_paper_image-5089/Visualization_t2m.png)
> > >
> > >
> > > [1] Y. Zhao et al., Image and Video Tokenization with Binary Spherical Quantization. In ICLR 2025.

---

### Decision · Program_Chairs · 2026-04-30

**Decision:**

Accept (spotlight)

**Comment:**

This paper proposes WLA (Weather Latent Autoencoder), a framework that compresses weather data into a unified latent space via a Pressure-Variable Unified Module (PVUM), VAEformer backbone, and Binary Quantization Module, enabling downstream tasks to operate at dramatically reduced storage and compute costs. Its main strengths include the novel hypernetwork-based PVUM for generalizing across pressure-variable subsets, strong compression performance, and practical downstream task evaluation. After the rebuttal, concerns related to computational costs, PVUM ablation, compression metric comparisons, generalization beyond temperature, and frequency-space analysis were largely resolved.